# Range-Net: A High Precision Neural SVD

## Abstract

For Big Data applications, computing a rank-$r$ Singular Value Decomposition (SVD) is restrictive due to the main memory requirements. Recently introduced streaming Randomized SVD schemes work under the restrictive assumption that the singular value spectrum of the data has an exponential decay. This is seldom true for any practical data. Further, the approximation errors in the singular vectors and values are high due to the randomized projection. We present Range-Net as a low memory alternative to rank-$r$ SVD that satisfies the lower bound on tail-energy given by Eckart-Young-Mirsky (EYM) theorem at machine precision. Range-Net is a deterministic two-stage neural optimization approach with random initialization, where the memory requirement depends explicitly on the feature dimension and desired rank, independent of the sample dimension. The data samples are read in a streaming manner with the network minimization problem converging to the desired rank-$r$ approximation. Range-Net is fully interpretable where all the network outputs and weights have a specific meaning. We provide theoretical guarantees that Range-Net extracted SVD factors satisfy EYM tail-energy lower bound with numerical experiments on real datasets at various scales that confirm these bounds. A comparison against the state-of-the-art streaming Randomized SVD shows that Range-Net is six orders of magnitude more accurate in terms of tail energy while correctly extracting the singular values and vectors.

## 1 Introduction

Singular Value Decomposition (SVD) is pivotal to exploratory data analysis in identifying an invariant structure under a minimalistic representation (assumptions on the structure) containing the span of resolvable information in the dataset. Finding a low rank structure is a fundamental task in applications including Image Compression (de Souza et al., 2015), Image Recovery (Brand, 2002), Background Removal (Wang et al., 2018), Recommendation Systems (Zhang et al., 2005) and as a pre-processing step for Clustering (Drineas et al., 2004) and Classification (Jing et al., 2017). With the advent of digital sensors and modern day data acquisition technologies, the sheer amount of data now requires that we revisit the solution scheme with reduced memory consumption as the target. In this work, we reformulate SVD with special emphasis on the main memory requirement, with no loss in accuracy, that precludes it's use for big data applications.

It is well known that natural data matrices have a decaying spectrum wherein saving the data in memory in its original form is either redundant or not required from an application point of view. However, any assumption on the decay rate can only be validated if the singular value decomposition is known *a priori*, which is seldom the case in exploratory data analysis (see **Fig. 3**). Visually assessing a rank-$r$ approximation for image processing applications might seem correct qualitatively (see **Fig. 4**) but are still prone to large errors due to limited human vision acuity (see **Fig. 6**). This is further exacerbated when the application at hand is associated with scientific computations wherein the anomalies or unaccounted phenomena are still being explored from large scale datasets. The reader is preemptively referred to **Fig. 19** where the high frequency features related to turbulence cannot be disregarded. Furthermore, for classification and clustering problems where feature dimension reduction is desirable it is imperative that a low-rank approximation of a dataset contains most ($\geq 90\%$) of the original information content without altering the subspace information. In this case, an over-sampled rank can exceed the feature dimension of the data itself (see **Section 4.2**).

## 1.1 PROBLEM STATEMENT

Let us denote the raw data matrix as $X \in \mathbb{R}^{m \times n}$ of rank $f \leq \min(m, n)$ and its approximation as $X_r \in \mathbb{R}^{m \times n}$. The SVD of $X$ is $X = U\Sigma V^T$, where $U \in \mathbb{R}^{m \times f} = [u_1, \cdots, u_f]$ and $V \in \mathbb{R}^{n \times f} = [v_1, \cdots, v_f]$ are its left and right singular vectors respectively, and $\Sigma \in \mathbb{R}^{f \times f} = diag(\sigma_1, \cdots, \sigma_f)$ are its largest non-zero singular values. The rank $r$ $(r \leq f)$ truncation of $X$ is then $X_r = U_r \Sigma_r V_r^T$, where $\Sigma_r = \Sigma_{[1:r]}$ are the top $r$ singular values, and $U_r = U_{[1:r]}$ and $V_r = V_{[1:r]}$ are the left and right singular vectors. In other words, $X = U\Sigma V^T = U_r \Sigma_r V_r^T + U_{f \setminus r} \Sigma_{f \setminus r} V_{f \setminus r}^T = X_r + X_{f \setminus r}$. Here, $U_{f \setminus r}, V_{f \setminus r}$ are the trailing $f - r$ left and right singular vectors.

**Theorem 1.** *Eckart-Young-Mirsky Theorem (Eckart & Young, 1936; Mirsky, 1960): Let $X \in \mathbb{R}^{m \times n}$ be a real, rank-f, matrix with $m \geq n$ with the singular value decomposition as $X = U\Sigma V^T$, where the orthonormal matrices $U, V$ contain the left and right singular vectors of $X$ and $\Sigma$ is a diagonal matrix of singular values. Then for an arbitrary rank-r, $r \leq f$ matrix $B_r \in \mathbb{R}^{m \times n}$,*

$$\|X - B_r\|_F \geq \|X - X_r\|_F$$

*where $X_r = U_r \Sigma_r V_r$ with $\Sigma_r$ is the diagonal matrix of the largest $r$ singular values and $U_r, V_r$ are the corresponding left and right singular vector matrices.*

The problem statement is then: Given $X \in \mathbb{R}^{m \times n}$ find $\hat{X}$ such that,

$$\underset{\hat{X} \in \mathbb{R}^{m \times n}, \ \text{rank}(\hat{X}) \leq r}{\arg \min} \|X - \hat{X}\|_F \tag{1}$$

In effect, the minimizer $\hat{X}_*$ of the above problem gives us the rank-$r$ approximation of $X$ such that $X_r = \hat{X}_*$. In this work we utilize the minimizer of the above problem to extract the top rank-$r$ SVD factors of $X$ without loading the entire data matrix into the main memory. *Note that the minimizer naturally gives the lower bound on this tail energy in addition to being a rank-r approximation.*

## 1.2 MAIN CONTRIBUTIONS

**Data and Representation Driven Neural SVD:** The representation driven network loss terms ensures that the data matrix $X$ is decomposed into the desired SVD factors such that $X = U\Sigma V^T$. In the absence of the representation enforcing loss term, the minimizer of Eq. 1 results in an arbitrary decomposition such that $X = ABC$ different from SVD factors.

**A Deterministic Approach with GPU Bit-precision Results:** The network can be initialized with weights drawn from a random distribution where the iterative minimization is deterministic. The streaming order of the samples is of no consequence and the user is free to choose the order in which the samples are processed in a batch-wise manner (indexed or randomized).

**First Streaming Architecture with Exact Low Memory Cost:** Range-net requires an exact memory specification based upon the desired rank-$r$ and data dimensions $X \in \mathbb{R}^{m \times n}$ given by $r(n + r)$, independent of the sample dimension $m$. This is the first streaming algorithm that does require the user to wait until the streaming step is complete, contrary to randomized streaming algorithms.

**Layer-wise Fully Interpretable:** Range-Net is a low-weight, fully interpretable, dense neural network where all the network weights and outputs have a precise definition. The network weights are placeholders for the right (or left) orthonormal vectors upon convergence of the network minimization problems (see **Appendix D**). The user can explicitly plug a ground truth solution to verify the network design and directly arrive at the tail energy bound.

## 2 RELATED WORKS

The core idea behind randomized matrix decomposition is to make one or more passes over the data and compute efficient sketches. They can be broadly categorized into four branches: 1) Sampling based methods (Subset Selection (Boutsidis et al., 2014) and Randomized CUR (Drineas et al., 2006)); 2) Random Projection based QR (Halko et al., 2011); 3) Randomized SVD (Halko et al., 2011); and 4) Power iteration methods (Musco & Musco, 2015). The sketches can represent any combination of row space, column space or the space generated by the intersection of rows and columns (core space). However, all of these methods require loading the entire data in memory. Readers are referred to Kishore Kumar & Schneider (2017); Ye et al. (2019) for an expanded survey.

Conventional SVD although deterministic and accurate, becomes expensive when the data size increases and requires $r$ passes over the data (see **Table 1**). The two branches of interest to us are the randomized SVD and Power iteration Methods for extracting SVD factors. Randomized SVD algorithms (Halko et al., 2011) are generally a two stage process: **1) Randomized Sketching** uses random sampling to obtain a reduced matri(x/ces) which covers any combination of the row, column and core space of the data; and **2) Deterministic Post-processing** performs conventional SVD on the reduced system from Randomized Sketching stage. These approaches make only one pass over the data assuming that the singular value spectrum decays rapidly.

Power iteration based approach (Musco & Musco, 2015) requires multiple passes over the data and are used when the singular value spectrum decays slowly. This class of algorithm constructs a Krylov matrix inspired by Block Lanczos (Golub & Underwood, 1977) to obtain a polynomial series expansion of the sketch. Although these algorithms achieve lower tail-energy errors, they cannot be used in big-data applications when $X$ itself is too large to be retained in the main memory. Here, constructing a Krylov matrix with higher order terms such as ($AA^T$ or $A^T A$) is not feasible[1].

Table 1: Current best randomized SVD Methods. $k, l, s$ are overestimated sketch sizes for a rank-$r$ estimate *s.t.* $k, l, s > r$. Note that Range-Net has an exact memory requirement (as conventional SVD), unlike order bounded Randomized methods. Note that deterministic implies the solution obtained upon convergence is deterministic.

| Method | Halko et al. (2011) | Upadhyay (2016) | Tropp et al. (2017b) | Tropp et al. (2019) | Range-Net | Conventional SVD |
|---|---|---|---|---|---|---|
| Space Complexity | $\mathcal{O}(k(m+n))$ | $\mathcal{O}(k(m+n)+s^2)$ | $\mathcal{O}(km+nl)$ | $\mathcal{O}(k(m+n)+s^2)$ | $r(n+r)$ | $n(m+2n)$ |
| # Passes | 1 | 1 | 1 | 1 | $\leq 5$ | r |
| Type | Randomized | Randomized | Randomized | Randomized | Deterministic | Deterministic |

Due to main-memory restrictions on remote compute machines, streaming (Clarkson & Woodruff, 2009; Liberty, 2013) algorithms became popular. For low-rank SVD approximations these involve streaming the data and updating low-memory sketches covering the row, column and core spaces. Existing randomized SVD capable of streaming include Halko et al. (2011); Upadhyay (2016); Tropp et al. (2017a; 2019), each with different sketch sizes and upper bounds on approximation errors (**Table 1**).

SketchySVD (Tropp et al., 2019) is the state of the art streaming randomized SVD, with sketch sizes comparable to it's predecessors and tighter upper bounds on the tail energy and lower errors. As a two stage approach, SketchySVD (**Alg. 1**) constructs an overestimated rank-$(k, s)$ sketch of the data based on row, column and core projections. A QR decomposition on the row and column sketches gives an estimate of the rank-$k$ subspace. This is followed by a conventional SVD on the core matrix to extract it's singular values and vectors. Finally, the singular vectors are returned after projecting them back to the original row and column space. The time cost of SketchySVD is $\mathcal{O}(k^2(m+n))$ with memory cost $\mathcal{O}(k(m+n)+s^2)$ with oversamling parameters $k = 4r+1$ and $s = 2k+1$.

### 2.1 LIMITATIONS OF RANDOMIZED SVD APPROACHES

We discuss a few limitations that led us to reformulate the problem in the spirit of EYM theorem. The reader is referred to **Appendix A** for a detailed discussion and supporting numerical examples to further enunciate these limitations.

**Tall and Skinny Matrices:** For a rank-$r$ approximation of $X \in \mathbb{R}^{m \times n}$ of rank-$f$, Randomized SVD methods rely upon rank-$k$ ($k > r$) sketches of $X$. However, these methods are useful only when $k \geq f$ but for practical datasets $f \leq \min(m, n)$ and therefore the memory requirement can still be overbearing. The reader is referred to **Section 4.2** for a practical example.

**Exponential Decay of Singular Values:** Assuming exponential decay implies the rank of $X$ itself is such that $f \ll \min(m, n)$. For real world applications, the data matrices are almost full rank $f \leq \min(m, n)$, where a rank $r$ truncation is chosen such that the desired dominant features are accounted for. **Appendix A** shows a synthetic case with non-exponential decay of singular values where sketching accrues substantial errors. Further, it is difficult to assume that the decay rate will follow a strict functional form: mixture of linear, exponential and others (see **Fig. 3**).

**Upper Bound on Tail Energy:** The problem statement in **Eq. 1** suggests finding a minimum with the minimizer providing a lower bound on the tail-energy. Even if the solution scheme is upper bounded (Halko et al., 2011; Upadhyay, 2016; Tropp et al., 2017a; 2019), the minimizer $\hat{X}_*$ in Eq. 1

---

[1] Readers are referred to **Fig. 5 (c)** and **Fig. 8** for a performance comparison between power iteration schemes and Range-Net for a mid-sized real and a synthetic dataset that can be loaded on our compute machine.

or equivalently achieving the lower bound is necessary.

**Approximation Errors:** A low-rank SVD solver that does not iteratively compute the projection (left or right) while solving **Eq. 1** cannot extract SVD factors with low errors even with multiple passes over the data matrix or multiple runs. As shown later in **Theorem 2**, any subspace projection (left or right) of the original data matrix that does not correspond to the minimizer in **Eq. 1** increases the tail energy and therefore results in incorrect low-rank SVD factors (singular values and vectors).

**Memory Requirement:** Randomized SVD requires an optimal choice of hyper-parameters (sketch sizes *etc.*) that are subjective to the dataset being processed. In a practical, limited memory scenario, this entails tuning the hyper-parameters for optimal trade-off between memory requirement, compute time and an approximation error that does not violate the upper bound on the tail energy.

**Remark.** *A low relative error in tail energy does not imply the extracted singular values and vectors will have similar relative errors at scale. The issue has been raised by Musco & Musco (2015), that merely upper bounding the tail energy equipped with Frobenius or spectral norm does not bound the approximation errors in the extracted singular values or vectors. Therefore, for a fair comparison we show error metrics on the extracted singular factors for all our numerical experiments.*

## 3    RANGE-NET: A 2-STAGE SVD SOLVER

We present **Range-Net** that explicitly relies upon solving the minimization problem in **Eq. 1** to achieve the lower bound on the tail-energy for a desired rank-$r$ approximation of a data matrix $X$ under a streaming setting. The readers are referred to **Appendix B** for the preliminaries followed by proofs of theorems and lemmas associated with each of the two stages.

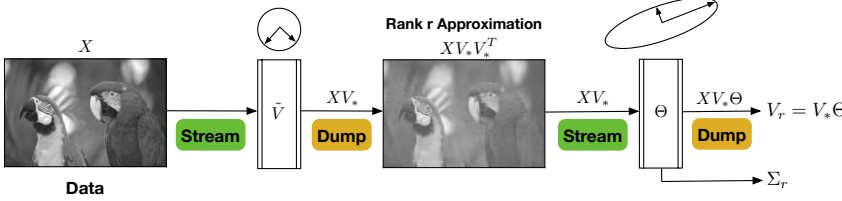

Figure 1: An overview of the low-memory, two-stage Range-Net SVD for Big Data Applications. **Stage 1** identifies the span of the desired rank-$r$ approximation. **Stage 2** rotates this span to align with the singular vectors while extracting the singular values of the data. The input data can be streamed from either a server or secondary memory. If the target is just a rank-$r$ compression then Stage-2 can be discarded without any loss in accuracy. Stage-2 only orders the rank-$r$ features based upon their respective tail energies.

### 3.1    NETWORK ARCHITECTURE

The proposed network architecture is divided into two stages: (1) Projection, and (2) Rotation, each containing only one dense layer of neurons and linear activation functions with no biases. **Fig. 2** shows an outline of the this two-stage network architecture where all the weights and outputs have a specific meaning enforced using representation and data driven loss terms. Contrary to randomized SVD algorithms the subspace projection (Stage 1) is not specified preemptively (consequently no assumptions) but is computed by solving an iterative minimization problem following EYM theorem corresponding to Eq. 1. The rotation stage (Stage 2) then reuses the EYM theorem again in a modified form to extract the singular vectors and values.

**Stage 1: Rank-$r$ Sub-space Identification:** The projection stage constructs an orthonormal basis that spans the $r$-dimensional sub-space of a data matrix $X \in \mathbb{R}^{m \times n}$ of an unknown rank $f \leq min(m, n)$. This orthonormal basis ($\tilde{V}$) is extracted as the stage-1 network weights once the network minimization problem converges to a fixed-point. The representation loss $\|\tilde{V}^T\tilde{V} - I_r\|_F$ in stage-1 enforces the orthonormal requirement on the projection space (even when $r > f$) while the data-driven loss $\|X - X\tilde{V}\tilde{V}^T\|_F$ minimizes the tail energy. Although the minimization problem is non-convex, the tail-energy is guaranteed to converge to the minimum at machine precision. The reader is referred to **Appendix C** for a discussion on the minimization problem (bi-quadratic loss function with $2^r$ global minima) for details regarding the convergence behavior.

**Theorem 2.** *For any $r, f \in \mathbb{Z}^+$, $0 < r \leq f$, if the tail energy of a rank-f matrix $X \in \mathbb{R}^{m \times n}$, $f \leq min(m, n)$, with respect to an arbitrary rank-r matrix $B_r = X\tilde{V}_r\tilde{V}_r^T$ is bounded below by the tail*

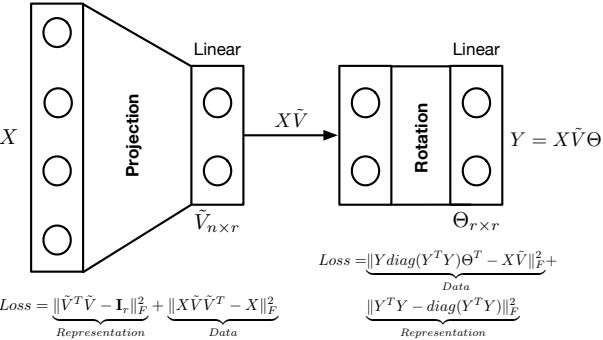

Figure 2: Network Architecture: Projection (Stage-1) and Rotation (Stage-2) for a 2-stage SVD

*energy of $X$ with respect to it's rank-$r$ approximation $X_r = XV_rV_r^T$ as, $\|X - B_r\|_F \geq \|X - X_r\|_F$ where, $V_r = span\{v_1, v_2, \cdots, v_r\}$ and $v_i$s are the right singular vectors corresponding to the largest $r$ singular values then the minimizer of $\arg\min_{\tilde{V} \in \mathbb{R}^{(n \times r)}} \|X - X\tilde{V}\tilde{V}^T\|_F$ is $V_*$ such that $V_*V_*^T = V_rV_r^T$.*

As per **Theorem 2**, the equality holds true only when $span\{B_r\} = span\{X_r\}$. Further, we define $B_r$ as, $B_r = X\tilde{V}\tilde{V}^T$ then, $V_*V_*^T = V_rV_r^T$ and $V_*^TV_* = I_r$ where $V_r$ is a rank-$r$ matrix with column vectors as top-r right singular vectors of $X$. The minimization problem then reads,

$$\min_{\tilde{V}} \quad \|X - X\tilde{V}\tilde{V}^T\|_F \quad s.t. \quad \tilde{V}^T\tilde{V} = I_r \tag{2}$$

with a minimum at the fixed point $V_* = span\{v_1, \ldots, v_r\}$ where $v_{i=1,\ldots,r}$ are the right singular vectors of $X_r$. This minimization problem describes the Stage 1 loss function of our network architecture. Upon convergence, the minimizer $\tilde{V}_*$ is such that $V_*V_*^T = V_rV_r^T$ following **Theorem 2** where $V_r$ is the matrix with columns as right singular vectors of $X$ corresponding to the largest $r$ singular values of $X$.

**Lemma 2.1.** *If $V_r^TV_r = I_r$ and $V_rV_r^T = V_*V_*^T$ then $V_*^TV_* = I_r$.*

**Lemma 2.2.** *If $X \in \mathbb{R}^{m \times n}$ is a rank $f$ matrix, then for any rank $r > f$, where $\{r, f\} \leq \min(m, n)$, if $V_*^TV_* = I_r$ and $V_*V_*^T = V_rV_r^T$ then $V_r^TV_r = I_r$.*

**Remark.** *Note that for $r \leq f$, the orthonormality constraint is trivially satisfied as shown in **Lemma 2.1**. However for $r > f$, the orthonormality constraint ensures that the column vectors in $V_*$ are orthonormal (see **Lemma 2.2**) allowing us to extract orthonormal right singular column vectors of $V_r$ from the Stage 2 minimization problem.*

**Stage 2: Singular Value and Vector Extraction:** The rotation stage then extracts the singular values by rotating the orthonormal vectors ($V_*$) to align with the right singular vectors ($V_r = V_*\Theta_r$). From the fixed point of the Stage-1 minimization problem Eq. 2 we have $V_*V_*^T = V_rV_r^T$. According to the EYM theorem the tail energy of a rank-$r$ matrix $XV_*C_r$, where $C_r$ is an arbitrary rank-$r$, real valued, square matrix, with respect to $XV_*$ is now bounded below by 0 or $\|XV_* - XV_*C_r\|_F \geq 0$.

**Theorem 3.** *Given a rank-$r$ matrix $XV_* \in \mathbb{R}^{m \times r}$ and an arbitrary, rank-$r$ matrix $C \in \mathbb{R}^{m \times r}$, following **Theorem 1**, the tail energy of $XV_*$ with respect to $XV_*C$ is bounded as, $\|XV_* - XV_*C\|_F \geq 0$, where the equality holds true if and only if $C = I_r$.*

**Lemma 3.1.** *If $C = \Theta_r\Theta_r^T$, where $\Theta_r \in \mathbb{R}^{r \times r}$ is a rank-$r$ matrix such that $C = I_r$, then $\Theta_r$ is a real-valued unitary matrix in an $r$-dimensional Euclidean space.*

**Theorem 4.** *Given a rank-$r$ matrix $XV_* \in \mathbb{R}^{m \times r}$, such that $V_*V_*^T = V_rV_r^T$ where $V_r$ is a matrix with column vectors as the top-$r$ right singular vectors of $X$, and a real-valued unitary matrix $\Theta_r \in \mathbb{R}^{r \times r}$ then $(XV_*\Theta_r)^T(XV_*\Theta_r)$ is a diagonal matrix $\Sigma_r^2$ where $\Sigma_r^2 = \text{diag}(\sigma_1^2, \sigma_2^2, \cdots, \sigma_r^2)$ and $\sigma_i$s are the top-$r$ singular values of $X$ if and only if $V_*\Theta_r = V_r$.*

From **Theorem 3** and **Lemma 3.1** we know that, $C_r = \Theta_r\Theta_r^T$, where $\Theta_r$ is a rank-$r$, unitary matrix in an $r$-dimensional Euclidean space. Further, from **Theorem 4** we have that $(XV_*\Theta_r)^T(XV_*\Theta_r)$ is a diagonal matrix $\Sigma_r^2 = \text{diag}(\sigma_1^2, \cdots, \sigma_r^2)$, where $\sigma_i$s are the top-$r$ singular values of $X$ if and only if $V_*\Theta_r = V_r$. Assuming $Y = XV_*\Theta_r$, the minimization problem now reads:

$$\min_{\Theta_r} \quad \|Y\Theta_r^T - XV_*\|_F \quad s.t. \quad Y^TY - \text{diag}(Y^TY) = 0 \tag{3}$$

**Remark.** *Note that stage 1 can be verified numerically independently of stage 2 by checking whether the orthonormality condition is met in addition to minimization problem converging to the tail-energy bound. Similarly, stage 2 minimization problem will return a rotation matrix $\Theta_r$ ($\Theta_r^T\Theta_r = \Theta_r\Theta_r^T = I_r$, $det(\Theta_r) = \pm 1$) upon convergence that can again be verified numerically.*

As discussed previously, this choice of loss terms equipped with a Frobenius norm ensures a rank-$r$ approximation in accord with the Eckart-Young-Mirsky (EYM) theorem. We therefore state that the expected values of the stage-1 loss term at the minimum must correspond to the rank $(n - r)$ tail energy. Further, the stage-2 loss is expected to reach a machine precision zero at the minimum. Once, the network minimization problems converge, the singular values are extracted from **Stage 2** network weights $\Theta_r$ as $\Sigma_r^2 = (XV_*\Theta_r)^T(XV_*\Theta_r)$. The right singular vectors can now be extracted using **Stage 2** layer weights given by $V_r = V_*\Theta_r$. Once $V_r$ and $\Sigma_r$ are known, left singular vectors $U_r = XV_*\Theta_r\Sigma_r^{-1}$. Please note that for $r > f$, $f - r$ singular values are zero and therefore $\Sigma_r^{-1}$ implies inverting the non-singular values using a threshold of $10^{-8}$.

**Implementation Details**: A detailed discussion and justification of our implementation choices are given in **Appendix E**. These include *a) Activation*: all the activation functions are linear with no biases, since SVD requires linearly separable orthogonal features; *b) Data Split*: we do not perform any split of the training data, since SVD factors are unique to the full data only; *c) Data Streaming*: we stream data from HDD into main memory to avoid data sample load; *d) Training and Setup*; *e)* Error Metrics; and *f)* Loss Profiles. Note that Range-Net has no hyper-parameters and therefore does not require any post-hoc tuning or adjustments.

## 4 RESULTS

We present numerical experiments for three datasets: (a) Parrot image, (b) MNIST, and (c) hurricane Sandy. The reader is referred to **Appendix E.5** for the definitions of the error metrics used. Note that, these error metrics rely upon conventional SVD as the baseline for a fair comparison. For additional numerical experiments on sparse graph datasets and other low rank approximations see **Appendix F**. SketchySVD's algorithmic implementation (**Alg. 1**) can be found in **Appendix G**.

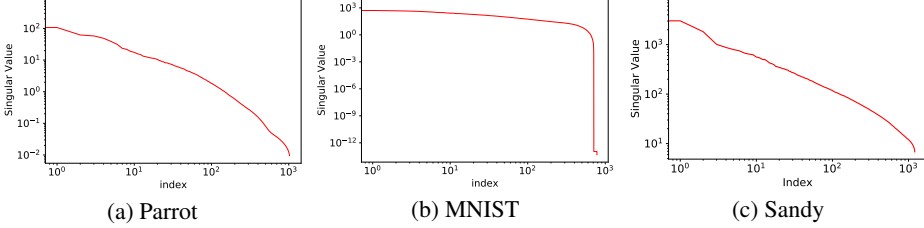

(a) Parrot                    (b) MNIST                    (c) Sandy

Figure 3: Singular value spectrum for the three practical datasets considered in this work. One can visually assess that the decay rate of the singular values is non-exponential.

### 4.1 IMAGE COMPRESSION: PARROTS (SVD)

As an example for SVD of natural images, we use the well known Parrots image from the image processing domain. The original image is in an RGB format, converted to a gray scale for demonstration purposes followed by normalization between $[0, 1]$. This $1024 \times 1536$ data matrix is then used to compute a rank $r = 20$ approximation for comparison and numerical analysis. **Fig. 4** shows the result of the low rank reconstruction for SVD, SketchySVD and Range-Net. Visually one can verify that **Fig. 4 (b), (d)** are similar while **Fig. 4 (c)** is different. To make the error in approximation more clear, we plot the absolute difference of SketchySVD and our net from the truncated rank image. **Fig. 4 (e), (f)** shows the the corresponding plots with heatmaps imposed for clarity. Notice that while the reconstruction error for our network ($\approx 10^{-7}$) is close to the GPU precision, SketchySVD has significantly higher error scale ($\approx 10^{-1}$), validating the artifacts in the approximated image.

The singular value spectrum does not decay exponentially (**Fig. 3**) and the data matrix is near-full rank ($f \approx 1024$). **Fig. 5 (a, b)** shows the scree error as the absolute difference between the predicted and the true singular values. For SketchySVD, the error fluctuates across the top $r = 20$ values, but also the scale of fluctuations is around $1$. Comparably, Range-Net incurs significantly lower errors in singular values (scale of $10^{-4}$). **Fig. 5 (c)** shows the reconstruction errors in Frobenius norm for SketchySVD (Tropp et al., 2019) (red line), Block Lanczos with Power Iteration (Musco & Musco, 2015) (black line), Sklearn's randomized SVD (skr) implementation Halko et al. (2011) with (solid

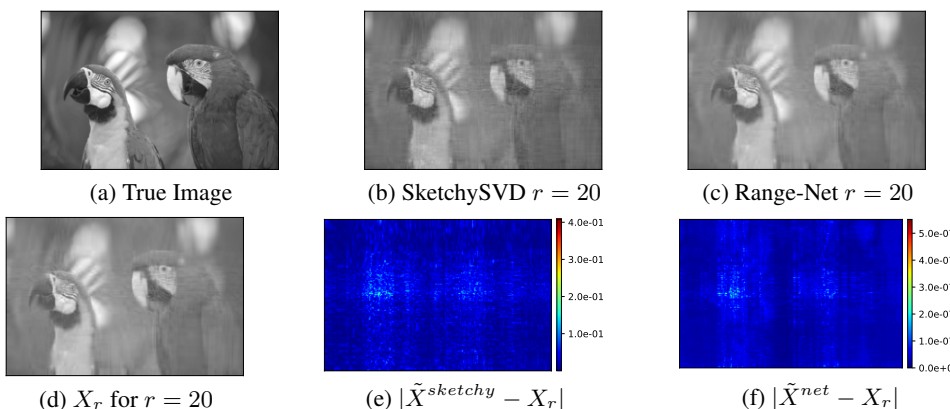

(a) True Image      (b) SketchySVD $r = 20$      (c) Range-Net $r = 20$

(d) $X_r$ for $r = 20$      (e) $|\tilde{X}^{sketchy} - X_r|$      (f) $|\tilde{X}^{net} - X_r|$

Figure 4: (a) True image, rank-20 reconstruction using (b) SktechySVD (oversampled rank $k = 81$) , (c) Range-Net (5-passes), (d) conventional SVD. Note that Sketchy SVD reconstruction error $(10^{-1})$ is 6 orders of magnitude apart from Range-Net's reconstruction error $(10^{-7})$ *w.r.t.* to the true $X_r$ from conventional SVD.

cyan line) and without power iteration (dashed blue line), and Range-Net (green line) over 1000 runs on the Parrot data. This shows that in order to gain lower reconstruction errors a power iteration is necessary that quickly becomes intangible in a big-data setting. Further, note that the expected error (upper bound) over multiple runs of Randomized SVD algorithms does not contract (reduce).

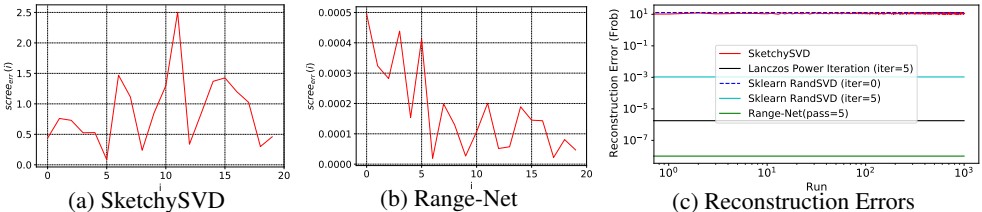

(a) SketchySVD      (b) Range-Net      (c) Reconstruction Errors

Figure 5: Scree error in the extracted singular values from (a) SketchySVD ($\approx 1$) and (b) Range-Net ($\approx 10^{-4}$). Notice the scale of errors. (c) Reconstruction errors (rank $r = 20$) for Range-Net and randomized SVD schemes (with and without power iterations) for Parrot image over 1000 runs.

**Fig. 6** shows the cross-correlation between extracted right singular vectors from SketchySVD (left) and Range-Net (right) against conventional SVD for a rank-20 approximation. SketchySVD oversampled rank is $k = 81$ and still the extracted right singular vectors deviate substantially. This implies that the extracted vectors from SketchySVD do not span the top rank-20 subspace of $X$ as opposed to Range-Net where stage-1 explicitly ensures this span without any oversampling. The higher the vector index, the higher the spread owing to random projections. Range-Net has a near-perfect cross-correlation with the true vectors, indicated by the solid diagonal and zero off-diagonal (near GPU-precision) entries.

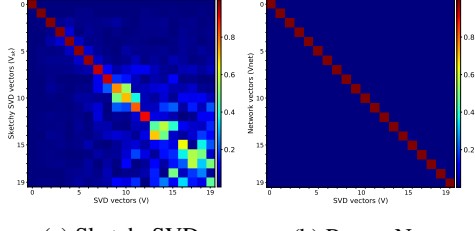

(a) SketchySVD      (b) Range-Net

Figure 6: Cross-correlation between true (conventional SVD) and extracted right singular vectors from (a) SketchySVD (b) Range-Net for a rank-$r = 20$ approximation of the Parrot image.

We tabulate error metrics (see **Appendix E** for definitions) for SketchySVD and Range-Net for various ranks in **Table. 2**. Note that all errors are reported *w.r.t.* the true SVD, where $err_{fr}$ and $err_{sp}$ denote the frobenius and spectral errors respectively. One can easily see that as the rank increases, SketchySVD's performance keeps on deteriorating for the $\chi^2_{err}$ metric (measure of correlation between true and estimated vectors), also evident from Fig. 6. Range-Net on the other hand has consistent lower errors in all metrics while simultaneously being memory efficient.

Table 2: Metric Performance and Memory of SketchySVD *vs.* Range-Net, for Parrot image

| Rank | SketchySVD | | | | | RangeNet | | | | |
|------|------------|--------|----------------|----------|----------|------------|----------|----------------|----------|----------|
| | $err_{fr}$ | $err_{sp}$ | $\chi^2_{err}$ | Mem (MB) | Time (s) | $err_{fr}$ | $err_{sp}$ | $\chi^2_{err}$ | Mem (MB) | Time (s) |
| $r = 10$ | 27.904 | 18.071 | 0.492 | 10.6 | 10 | 0.0 | 0.0 | 0.018 | 0.33 | 21 |
| $r = 20$ | 11.974 | 2.201 | 0.662 | 21.69 | 14 | 0 | 0 | 0.023 | 0.69 | 26 |
| $r = 50$ | 2.772 | 0.181 | 0.762 | 59.87 | 30 | 1.91e-7 | 0 | 0.027 | 1.72 | 43 |
| $r = 100$ | 0.614 | 2.14e-2 | 0.923 | 139.9 | 63 | 2.32e-7 | 1.08e-7 | 0.033 | 3.6 | 72 |

## 4.2 Dimension Reduction: MNIST (Eigen / PCA)

Principal Component Analysis (PCA) is a special variant of Eigen decomposition, where the samples are mean corrected before constructing a feature covariance matrix followed by Eigen decomposition. Note that Range-Net does not require construction of the feature covariance matrix and can directly extract the eigenvectors and values without any modification. This is due to the fact that the Eigenvalues are the square of the singular values for any non-square data matrix $X \in \mathbb{R}^{m \times n}$ where right singular vectors are exactly the same as the eigenvectors.

MNIST has 60k images of size $28 \times 28$ in the training set. We reshape each image into a 784-dim vector to obtain the data matrix $X \in \mathbb{R}^{60000 \times 784}$, as a *tall skinny matrix*. In a streaming setting, the mean feature vector computation requires one pass over the data matrix. This can be subsequently used during the network training (Stage-1) to mean correct streamed input vectors.

Table 3: Metric Performance and Memory of SketchySVD *vs.* Range-Net, for MNIST

| Rank | SketchySVD | | | | | RangeNet | | | | |
|---|---|---|---|---|---|---|---|---|---|---|
| | $err_{fr}$ | $err_{sp}$ | $\chi^2_{err}$ | Mem (GB) | Time (s) | $err_{fr}$ | $err_{sp}$ | $\chi^2_{err}$ | Mem (MB) | Time (s) |
| $r = 20$ | 1.09e3 | 8.29e2 | 1.34 | 0.47 | 216 | 0 | 0 | 0.012 | 0.51 | 416 |
| $r = 50$ | 1.03e3 | 8.25e2 | 2.14 | 1.18 | 384 | 0 | 0 | 0.025 | 1.33 | 552 |
| $r = 100$ | 1.05e3 | 8.47e2 | 2.03 | 2.38 | 702 | 1.12e-7 | 1.01e-7 | 0.052 | 2.83 | 776 |
| $r = 200$ | 1.14e3 | 8.62e2 | 1.0 | 4.84 | 1452 | 2.36e-7 | 2.52e-7 | 0.071 | 6.29 | 1256 |

For this dataset, it is well known that $r = 200$ captures $\geq 90\%$ variance in the dataset. For SketchySVD, this results in projection matrices of ranks $k = 4r + 1 = 801$ and $s = 2k + 1 = 1602$. Since MNIST only has $n = 784$ features, SketchySVD (**Alg. 1**) is almost equal if not more memory intensive than conventional SVD for such tall and skinny matrices. **Table. 3** shows the error metrics under different rank setting, where even with oversampling, SketchySVD's errors are high. Range-Net on the other hand with an exact memory requirement $(r(n + r))$ can handle much larger full rank tall and skinny matrices without incurring extraneous memory load. As discussed before, since this data matrix is tall and skinny ($60k \times 784$) we already know that for SketchySVD any rank-$r$ *s.t.* ($r \geq 196$) will result in the oversampling parameters $k \geq 784$ and $s \geq 1569$. SketchySVD will now extract lower-error SVD factors since the oversampled rank redundantly exceeds the feature dimension.

## 4.3 Scientific Computing: Sandy Big Data (SVD)

Satellite data gathered by NASA for Hurricane Sandy over the Atlantic ocean represents the big data counter-part for scientific computations. The data-set is openly available[2] and comprises of RGB snapshots captured at approximately one-minute interval. The full data-set consists of $896 \times 719$ pixel images for 1208 time-instances is of size 24 **GB**. We chose this particular big data so that a conventional SVD can be performed on our machine (16 GB RAM) for benchmarking. Please note that this restriction is imposed by conventional SVD method due to its high main memory requirements. In contrast, our neural SVD solver can handle data sets that are orders of magnitude larger in size with the same hardware specification.

Range-Net can not only handle larger datasets than SketchySVD, but also ensures lower errors in approximating the SVD factors. **Tab. 4** shows the error metrics for Range-Net with a comparison of peak main-memory load between SketchySVD and Range-Net for ranks $r = [10, 50, 100]$. **Appendix F.3** shows a comparison of dynamic mode reconstructions obtained from SketchySVD, conventional full rank SVD, and Range-Net and the associated scree errors in the computed singular values.

Table 4: Metric Performance and Memory of SketchySVD *vs.* Range-Net, for Sandy

| Rank | SketchySVD | | | | | RangeNet | | | | |
|---|---|---|---|---|---|---|---|---|---|---|
| | $err_{fr}$ | $err_{sp}$ | $\chi^2_{err}$ | Mem (GB) | Time (s) | $err_{fr}$ | $err_{sp}$ | $\chi^2_{err}$ | Mem (MB) | Time (s) |
| $r = 10$ | 1.43e3 | 7.72e2 | 0.47 | 2.56 | 325 | 0 | 0 | 0.011 | 0.39 | 371 |
| $r = 50$ | 7.18e2 | 1.81e2 | 2.04 | 12.48 | 507 | 0 | 0 | 0.018 | 2.01 | 607 |
| $r = 100$ | 4.32e2 | 6.68e1 | 3.022 | 24.91 | 792 | 1.12e-7 | 1.24e-7 | 0.021 | 4.19 | 779 |

Randomized SVD extracted factors deviate quite substantially when the user specified rank $r$ is such that the oversampled rank $k$ is much lower than the unknown rank $f$ of a given data matrix. The reader is also referred to the additional experiment in **Appendix F.4** where a low rank ($r = 10$) approximation is extracted. Here, SketchySVD deviates quite substantially after rank-3 while Range-Net still remains in excellent agreement with the baseline singular vectors and values.

---

[2]https://www.nasa.gov/mission_pages/hurricanes/archives/2012/h2012_Sandy.html

### 4.4 STORAGE COMPLEXITY ANALYSIS

To estimate the memory efficiency of Range-Net, let us consider the peak main memory (RAM) requirement for the compuation of SVD factors. Range-Net has two layers in succession, one corresponding to the low-rank projector $\tilde{V}^{r \times n}$ and the rotation matrix $\Theta^{r \times r}$. For Sketchy SVD, the peak memory load occurs during the construction of a core matrix $C^{s \times s}$ (see **Alg. 1**). This requires that the two projection matrices $\Phi^{m \times s}, \Psi^{n \times s}$, one projected data matrix $Z^{s \times s}$, and two rank-$k$ decomposition $Q^{m \times k}, P^{n \times k}$ and the core matrix $C^{s \times s}$, be present in the memory simultaneously. The memory efficiency factor $(s_{eff})$ for a rank-$r$ approximation with $k = 4r + 1, s = 2k + 1$ is:

$$s_{eff} = \frac{\text{SketchySVD}}{\text{Range-Net}} = \frac{ns + ms + 2s^2 + mk + nk}{rn + r^2} = \frac{(m+n)(k+s) + 2s^2}{r(n+r)} \approx \frac{12(m+n) + 128r}{(n+r)}$$

$$\approx 7.67e2 \quad \text{for MNIST}(m = 60k, n = 784, r = 200)$$

To validate the ratio, we constructed a synthetic dataset of $m = 50k$ rows and the number of columns were varied starting at $n = 10k$ with increments of $10k$. The expected rank was held at $r = 200$. **Fig. 7** shows the memory allocation (in MegaBytes (MB)) of SketchySVD *vs.* Range-Net, while $n$ varies between $[10k - 150k]$. When $m = 50k, n = 150k$ and $r = 200$, SketchySVD has a peak memory consumption of 14GB due to oversampling parameters of $k = 801, s = 1603$, while Range-

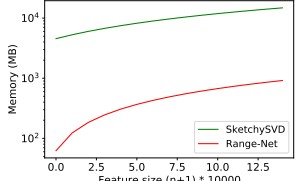

Figure 7: Peak memory load of Range-Net *vs.* SketchySVD on synthetic data.

Net only requires 916MB. Since Range-Net has an exact memory requirement of $r(n + r)$ for a rank-$r$ SVD, it always occupies main lesser memory than oversampling of SketchySVD (or any other randomized method) by two orders of magnitude, and scalable to larger datasets.

## 5 DISCUSSION

From the numerical experiments it is evident that Range-Net performs at par with conventional SVD with GPU bit precision results, while being extremely memory efficient. Range-Net constructs the rank-$r$ projector $VV^T$ iteratively to reach the tail energy lower bound given by EYM. Alternatively, the rank-$r$ minimizer $V$ of **Eq. 1** gives the correct rank-$r$ right projector $VV^T$ of $X$. Any arbitrary (random) projection of $X$ onto an oversampled rank-$k$ subspace $(k > r)$, rather than the tail energy minimizing subspace $VV^T$, can inadvertently annihilate the desired top ranking singular features resulting in irreducible approximation errors.

This issue is specially important in exploratory data analysis for scientific computing, where one is not only interested in the top-most singular values of the dataset but also the dominant phenomena (singular vectors). Furthermore, with increasing digital sensor resolution the focus now is to isolate and study the lower energy (high frequency/ small spatial scale) features as in the case of Hurricane Sandy, where turbulence manifests as low rank features (see **Figs. 19, 24** in Appendix).

We again point out that accuracy is gained by achieving (or finding) this minimizer or lower bound of the rank-$r$ tail-energy of $X$ and therefore the upper bound on this tail-energy is of no consequence to Range-Net. As pointed out by Musco & Musco (2015), upper bounding the tail energy does not ensure that the approximation errors in the extracted singular values or vectors will reduce. Therefore, small relative tail energy errors can mislead the reader that the singular factors are equivalently accurate wherein the absolute errors can still be substantially large.

## 6 CONCLUSION

We present Range-Net as a low-weight, high-precision, fully interpretable neural SVD solver for big data applications that is independently verifiable without performing a full SVD. We show that our solution approach achieves lower errors metrics for the extracted singular vectors and values compared to Randomized SVD methods. A discussion is also provided on the limiting assumptions and practical consequences of using Randomized SVD schemes for big data applications. We also verify that our network minimization problems converges to the EYM tail energy bound in Frobenius norm at machine precision. A number of practical problems are considered, where SVD or Eigen decompositions are required, that demonstrate the applicability of Range-Net to large scale datasets. A fair comparison is provided against the state of the art randomized, streaming SVD algorithm with conventional SVD solution as the baseline for benchmarking and verification.

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

## A  NEED FOR RANGE-NET

To illustrate the limitations of current streaming randomized SVD approaches, we consider a synthetic data matrix $X$ with slow decay in the singular value. The numerical results section later presents a number of these singular value spectra for different practical datasets (**Fig. 3**) to demonstrate that the decay rates are subjective to the problem at hand.

$$X = \text{diag}(\underbrace{450, 449, \cdots, 2, 1}_{f=450}, \underbrace{0, \cdots, 0}_{n-f})$$

Here $X \in \mathbb{R}^{m \times n}$ is a strictly diagonal matrix with $m = n = 500$ with rank $f = 450$, where the singular value spectrum decays linearly. **Fig. 8** shows a comparison of reconstruction errors (see Metrics in **Appendix E.5** for the definition) for SketchySVD (Tropp et al., 2019) (red line), Block Lanczos with Power Iteration (Musco & Musco, 2015) (black line), Sklearn's randomized SVD (skr) implementation (Halko et al., 2011) with (solid cyan line) and without power iteration (dashed blue line), and Range-Net (green line) over 1000 runs for this synthetic dataset.

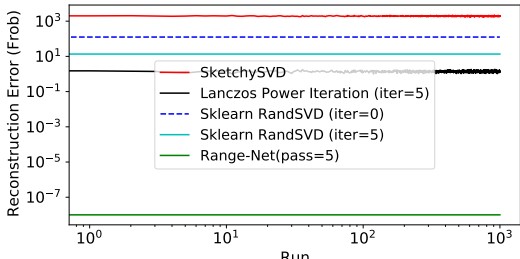

Figure 8: Reconstruction errors ($r = 20$) for Range-Net and randomized SVD schemes (with and without power iterations) for the non-exponentially decaying singular value spectrum over 1000 runs.

Please note that although power iteration improves the reconstruction error for both Block Lanczos (Musco & Musco, 2015) and Sklearn's RandSVD (Halko et al., 2011), power iteration itself requires a persistent presence of the data matrix $X$ in the main memory. For a practical big data scenario, power iteration is therefore not a feasible alternative when the data matrix $X$ or it's sketch is itself too big to be loaded into the main memory. *Note that the error expectation (upper bound) over multiple runs of Randomized SVD algorithms does not reduce.* We further identify the following requirements for SketchySVD to return SVD factors with relatively lower approximation errors:

1. Decay rate of singular values of a dataset must be exponential.

2. For a rank-$f$ matrix, the desired rank $r$ must be chosen such that the oversampled rank $k$ is strictly greater than $f$ ($k \geq f$) to achieve lower errors at scale compared to other runs.

We suggest that the reader also attempt the case where all the diagonal entries are strictly ones and zeros under a high rank setting.

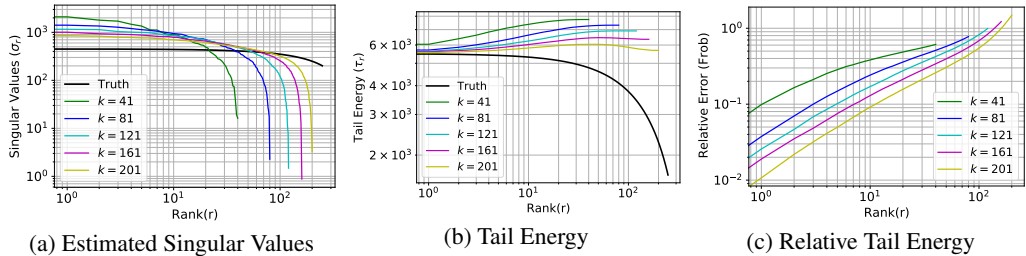

(a) Estimated Singular Values          (b) Tail Energy          (c) Relative Tail Energy

Figure 9: SketchySVD approximation errors for a synthetic dataset with linear decay in singular value spectrum, corresponding to $r = [10, 20, 30, 40, 50]$ with corresponding oversampled rank $k = [41, 81, 121, 161, 201]$. Since the decay is non-exponential, SketchySVD accrues large approximation errors, hence impractical for real datasets with similar behavior.

**Fig. 9** shows the singular values extracted by SketchySVD for a linearly decaying spectrum with corresponding errors in absolute and relative tail energies. The reader is referred to **Appendix**

**B.1** for the definitions of tail energy and relative tail energy. Note that the synthetic data is a diagonal matrix chosen specifically so that the exact tail energies can be computed using Frobenius norm as $\left(\sum_{i=r+1}^{k} x_{ii}^2\right)^{\frac{1}{2}}$. For a rank-$r$ approximation, SketchySVD suggests oversampling by a factor of $k = 4r + 1$ to extract the rank-$r$ factors correctly. Hence for an oversampled rank $k = [41, 81, 121, 161, 201]$ the corresponding top rank $r = [10, 20, 30, 40, 50]$ extracted singular values and vectors will have the lowest approximation errors. However as shown in the **Fig. 9 (a)** the extracted singular values have an order of magnitude difference *w.r.t.* the ground truth. Consequently, **Fig. 9 (b)** shows that the absolute tail-energies of the extracted features deviate quite substantially from the true tail-energy. Furthermore, we also notice that the deviations remains large as long as the oversampled rank-$k$ is such that $k < f$. For a practical dataset $f$ is either unknown or almost full rank $f \leq \min(m, n)$ or both and can only be detected by performing a full SVD on the dataset. This poses a serious restriction on SketchySVD's reliability for a realistic big data application, due to an exponential decay assumption.

We also notice that for smaller values of $r$, the accrued error in both the extracted singular values and tail energy error is worse. **Fig. 9 (b)** shows that for different rank approximations SketchySVD tail energy deviates from the truth quite substantially. This is due to the fact that the oversampled rank $k = 4r + 1 \ll f$, as pointed out before. Note that for oversampling parameter $k < f$, although the error decreases as $k \to f$ the memory requirement increases correspondingly as $\mathcal{O}(k)$ for extracting a low rank-$r$ approximation. This implies that for slow decaying spectrum optimal values of $k$ are such that $k \geq f$ even when a rank-1 approximation is desired. In **Fig. 9 (c)**, one can observe *relative errors* between $10^{-2}$ and $10^0$. Although this implies that the actual rank-$r$ tail-energy approximation error is off by $1\%$ at the best, the extracted singular values and vectors are off by one order of magnitude. As a consequence, the extracted singular vectors no longer represent the features of the dataset.

**Remark.** *For a randomized SVD algorithm to converge (without power iterations) to a rank-$r$ approximation $X_r$ over multiple runs, we posit that a rank-$r$ sketch matrix $\hat{X}$ for a given rank-$f$ dataset $X$, for $f \geq r$, be such that $P(span\{\hat{X}\} \cap span\{X_r\} = span\{X_r\}) \geq 0.5$. However, ensuring this requires substantial amount of prior knowledge or intelligent sampling (a multipass iterative algorithm).*

Range-Net with it's explicit minimization of tail energy is capable of intelligent sampling on an arbitrary matrix without requiring any prior information. The key point to note here is that Range-Net relies upon an iterative computation of a near optimal projector instead of arbitrary/user-specified projectors used in Randomized SVD schemes. Even if the tail energy is theoretically upper bounded for some of the Randomized SVD schemes, the target is to find the lower-bound (minimizer) on the tail energy as discussed in **Section 1.1**. Furthermore, since none of the randomized SVD schemes construct the projector in an iterative manner while minimizing Eq. 1, the relative error in the tail energy remains high. Even if multiple runs of SketchySVD or Sklearn's RandSVD are performed, the reconstruction errors in tail energies remain the same at scales shown in **Fig. 8**. We would also like to point out that although one must strive for lower errors (relative or otherwise) and tighter theoretical upper and lower bounds, in practice we should also closely monitor if these theoretical bounds deliver us the desired solution.

# B   THEORETICAL GUARANTEES

## B.1   PRELIMINARIES

The Frobenius norm of a matrix $A$ is given by,

$$\|A\|_F = \left(\sum_i \sum_j a_{ij}^2\right)^{\frac{1}{2}} = \left(\text{Tr}(A^T A)\right)^{\frac{1}{2}} = \left(\text{Tr}(AA^T)\right)^{\frac{1}{2}}$$

Further, the Frobenius norm can be used to bound Trefethen & Bau III (1997) the norm of a matrix product as,

$$\|AB\|_F \leq \|A\|_F \|B\|_F$$

For a Frobenius norm we have that,

$$\|A + B\|_F \leq \|A\|_F + \|B\|_F$$
$$\|A - B\|_F \geq \big|\|A\|_F - \|B\|_F\big|$$

Also the Frobenius norm of a rank-$f$ matrix $A$,

$$\|A\|_F = \|\Sigma_f\|_F$$

where $\Sigma_f = \mathrm{diag}(\sigma_1, \sigma_2, \cdots, \sigma_f)$ and $\sigma_i$s are the $f$ non-zero, singular values of $A$.

Let $A, B, C$ be matrices such that the following matrix products are feasible. The cyclic property of the linear trace operator is,

$$\mathrm{Tr}(ABC) = \mathrm{Tr}(BCA) = \mathrm{Tr}(CAB)$$

**Definition 1.** *The tail energy of an arbitrary matrix $B \in \mathbb{R}^{m \times n}$ with respect to a given matrix $X \in \mathbb{R}^{m \times n}$ equipped with a Frobenius norm is defined as,*

$$\tau = \|X - B\|_F$$

**Definition 2.** *Let $r, f \in \mathbb{Z}^+$ be positive integers such that $0 < r \leq f$, then a rank-$r$ truncation $X_r$ of a rank-$f$ matrix $X$ is defined as,*

$$X_r = U_r \Sigma_r V_r^T = X V_r V_r^T$$

*where, $\Sigma_r = \mathrm{diag}(\sigma_1, \sigma_2, \cdots, \sigma_r)$ and $\sigma_i$s are the top $r$ singular values of $X$ and $V_r = [v_1, v_2, \cdots, v_r]$, and $U_r = [u_1, u_2, \cdots, u_r]$ are matrices such that $v_i$s and $u_i$s are the corresponding right and left singular vectors, respectively.*

The relative tail energy of a rank-$r$ matrix $B_r$ with respect to a rank-$f$ matrix $X$ ($r \leq f$) is then defined as,

$$\tau_{rel,r} = \frac{\|X - B_r\|_F}{\|X - X_r\|_F} - 1$$

### B.2 STAGE 1

**Proof of Theorem 2.** *For any $r, f \in \mathbb{Z}^+$, $0 < r \leq f$, if the tail energy of a rank-$f$ matrix $X \in \mathbb{R}^{m \times n}$, $f \leq min(m, n)$, with respect to an arbitrary rank-$r$ matrix $B_r = X \tilde{V}_r \tilde{V}_r^T$ is bounded below by the tail energy of $X$ with respect to it's rank-$r$ approximation $X_r = X V_r V_r^T$ as,*

$$\|X - B_r\|_F \geq \|X - X_r\|_F$$

*where, $V_r = span\{v_1, v_2, \cdots, v_r\}$ and $v_i$s are the right singular vectors corresponding to the largest $r$ singular values then the minimizer of $\underset{\tilde{V} \in \mathbb{R}^{(n \times r)}}{\arg\min} \|X - X \tilde{V} \tilde{V}^T\|_F$ is $V_*$ such that $V_* V_*^T = V_r V_r^T$.*

From **Theorem 1** we have,

$$\|X - B_r\|_F - \|X - X_r\|_F \geq 0 \tag{4}$$

Let $V_r = \{v_1, v_2, \ldots, v_r\}$ be the top-r, right-singular vectors of $X$ corresponding to the largest singular values,

$$X_r = X V_r V_r^T = U \Sigma V^T V_r V_r^T = U \Sigma V_r^T \tag{5}$$

Also let $B_r = X \tilde{V}_r \tilde{V}_r^T$ where $\tilde{V}_r$ is an arbitrary rank-$r$ matrix. From triangle inequality we have that,

$$\|X(V_r V_r^T - \tilde{V}_r \tilde{V}_r^T)\|_F \geq \|X(I_n - \tilde{V}_r \tilde{V}_r^T)\|_F - \|X(I_n - V_r V_r^T)\|_F \tag{6}$$

Combining Eq. 4 and Eq. 6 we get,

$$\|X(V_r V_r^T - \tilde{V}_r \tilde{V}_r^T)\|_F \geq 0 \tag{7}$$

Additionally,

$$\|X\|_F \|V_r V_r^T - \tilde{V}_r \tilde{V}_r^T\|_F \geq \|X(V_r V_r^T - \tilde{V}_r \tilde{V}_r^T)\|_F \tag{8}$$

Using the above two inequalities we arrive at,

$$\|X\|_F \|V_r V_r^T - \tilde{V}_r \tilde{V}_r^T\|_F \geq 0 \tag{9}$$

Since $\|X\|_F > 0$, equality is achieved when $\tilde{V}_r \tilde{V}_r^T = V_* V_*^T = V_r V_r^T$. In other words, $span\{V_r\} = span\{V_*\}$ since $(V_* \Theta_r)(V_* \Theta_r)^T = V_* V_*^T$ for any rank-$r$, real valued, unitary matrix $\Theta_r$ spanning the top rank-$r$ subspace of $X$.

**Remark.** *Theorem 2 also implies that any matrix $\tilde{V}_r \tilde{V}_r^T$ that does not span the same rank-r subspace of $X$ as $V_r V_r^T$ will result in a higher tail-energy than given by the EYM tail-energy bound equipped with a Frobenius norm.*

**Proof of Lemma 2.1.** *If $V_r^T V_r = I_r$ and $V_r V_r^T = V_* V_*^T$ then $V_*^T V_* = I_r$.*

Let us assume $V_r^T V_r = I_r$ then,

$$\|V_r^T V_r - I_r\|_F^2 = 0$$
$$\mathrm{Tr}\left(V_r^T V_r V_r^T V_r\right) + \mathrm{Tr}\left(I_r\right) - 2\,\mathrm{Tr}\left(V_r^T V_r\right) = 0$$

Using the cyclic property of the trace operator we have,

$$\mathrm{Tr}\left(V_r V_r^T V_r V_r^T\right) + \mathrm{Tr}\left(I_r\right) - 2\,\mathrm{Tr}\left(V_r V_r^T\right) = 0$$

Using **Theorem 2**, $V_* V_*^T = V_r V_r^T$,

$$\mathrm{Tr}\left(V_* V_*^T V_* V_*^T\right) + \mathrm{Tr}\left(I_r\right) - 2\,\mathrm{Tr}\left(V_* V_*^T\right) = 0$$

Again using the cyclic property of the trace operator we now get,

$$\mathrm{Tr}\left(V_*^T V_* V_*^T V_*\right) + \mathrm{Tr}\left(I_r\right) - 2\,\mathrm{Tr}\left(V_*^T V_*\right) = 0$$

Hence, $V_*^T V_* = I_r$. This shows that following **Theorem 2**, the matrix $V_*$ comprises of orthonormal column vectors spanning the same top rank-$r$ subspace of $X$ as the orthonormal column vectors $V_r$.

**Proof of Lemma 2.2.** *If $X \in \mathbb{R}^{m \times n}$ is a rank $f$ matrix, then for any rank $r > f$, where $\{r, f\} \leq \min(m, n)$, if $V_*^T V_* = I_r$ and $V_* V_*^T = V_r V_r^T$ then $V_r^T V_r = I_r$.*

$$\|V_*^T V_* - I_r\|_F^2 = 0$$
$$\mathrm{Tr}\left(V_*^T V_* V_*^T V_*\right) + \mathrm{Tr}\left(I_r\right) - 2\,\mathrm{Tr}\left(V_*^T V_*\right) = 0$$

Using the cyclic property of the trace operator we have,

$$\mathrm{Tr}\left(V_* V_*^T V_* V_*^T\right) + \mathrm{Tr}\left(I_r\right) - 2\,\mathrm{Tr}\left(V_* V_*^T\right) = 0$$

Using **Theorem 2**, $V_* V_*^T = V_r V_r^T$,

$$\mathrm{Tr}\left(V_r V_r^T V_r V_r^T\right) + \mathrm{Tr}\left(I_r\right) - 2\,\mathrm{Tr}\left(V_r V_r^T\right) = 0$$

Again using the cyclic property of the trace operator,

$$\mathrm{Tr}\left(V_r^T V_r V_r^T V_r\right) + \mathrm{Tr}\left(I_r\right) - 2\,\mathrm{Tr}\left(V_r^T V_r\right) = 0$$
$$\mathrm{Tr}\left(V_r^T V_r V_r^T V_r + I_r - 2V_r^T V_r\right) = 0$$
$$\|V_r^T V_r - I_r\|_F = 0$$

Hence $V_r^T V_r = I_r$.

**Remark.** *Lemma 2.2 shows that for a rank-r approximation of a rank-f matrix $X$ such that $r > f$, the extracted right singular vectors are orthonormal when $V_*^T V_* = I_r$. This justifies the constraint $\tilde{V}^T \tilde{V} = I_r$ for the stage-1 minimization problem in Eq. 2 and is numerically verified in **Appendix D**.*

**Remark.** *Note that Lemma 2.1 and 2.2 does not imply that $V_* = V_r$ instead $V_* \Theta_r = V_r$, where $\Theta_r$ is any real valued unitary matrix for the equality to hold true.*

### B.3 STAGE 2

**Proof of Theorem 3.** *Given a rank-r matrix $XV_* \in \mathbb{R}^{m \times r}$ and an arbitrary, rank-r matrix $C \in \mathbb{R}^{m \times r}$, following **Theorem 1**, the tail energy of $XV_*$ with respect to $XV_*C$ is bounded as,*

$$\|XV_* - XV_*C\|_F \geq 0$$

*where the equality holds true if and only if $C = I_r$.*

$$\|XV_*(I_r - C)\|_F \geq 0$$
$$\|XV_*\|_F \|I_r - C\|_F \geq \|XV_*(I_r - C)\|_F$$
$$\|XV_*\|_F \|I_r - C\|_F \geq 0$$

Since $XV_* > 0$, this implies equality is achieved if and only if $C = I_r$.

**Proof of Lemma 3.1** *If $C = \Theta_r \Theta_r^T$, where $\Theta_r \in \mathbb{R}^{r \times r}$ is a rank-r matrix such that $C = I_r$, then $\Theta_r$ is a real-valued unitary matrix in an r-dimensional Euclidean space.*

$$\Theta_r \Theta_r^T = I_r$$
$$\|\Theta_r \Theta_r^T - I_r\|_F^2 = 0$$
$$\text{Tr}\left(\Theta_r \Theta_r^T \Theta_r \Theta_r^T + I_r - 2\Theta_r \Theta_r^T\right) = 0$$
$$\text{Tr}\left(\Theta_r \Theta_r^T \Theta_r \Theta_r^T\right) + \text{Tr}\left(I_r\right) - 2\text{Tr}\left(\Theta_r \Theta_r^T\right) = 0$$

Using the cyclic property of the trace operator,

$$\text{Tr}\left(\Theta_r^T \Theta_r \Theta_r^T \Theta_r\right) + \text{Tr}\left(I_r\right) - 2\text{Tr}\left(\Theta_r^T \Theta_r\right) = 0$$
$$\text{Tr}\left((\Theta_r^T \Theta_r - I_r)(\Theta_r^T \Theta_r - I_r)^T\right) = 0$$
$$\|\Theta_r^T \Theta_r - I_r\|_F^2 = 0$$

This implies $\Theta_r^T \Theta_r = I_r$. Since $\Theta_r^T \Theta_r = \Theta_r \Theta_r^T = I_r$ this implies that $\Theta_r$ is a real-valued unitary matrix in the $r$-dimensional Euclidean space.

**Proof of Theorem 4.** *Given a rank-r matrix $XV_* \in \mathbb{R}^{m \times r}$, such that $V_* V_*^T = V_r V_r^T$ where $V_r$ is a matrix with column vectors as the top-r right singular vectors of $X$, and a real-valued unitary matrix $\Theta_r \in \mathbb{R}^{r \times r}$ then $(XV_* \Theta_r)^T(XV_* \Theta_r)$ is a diagonal matrix $\Sigma_r^2$ where $\Sigma_r^2 = \text{diag}(\sigma_1^2, \sigma_2^2, \cdots, \sigma_r^2)$ and $\sigma_i s$ are the top-r singular values of $X$ if and only if $V_* \Theta_r = V_r$.*

$$(XV_* \Theta_r)^T(XV_* \Theta_r) = \Sigma_r^2$$
$$\Theta_r^T V_*^T (X^T X) V_* \Theta_r = \Sigma_r^2$$
$$V_*^T X^T X V_* = \Theta_r \Sigma_r^2 \Theta_r^T$$
$$V_* V_*^T X^T X V_* V_*^T = V_* \Theta_r \Sigma_r^2 \Theta_r^T V_*^T$$

Using $V_* V_*^T = V_r V_r^T$ from **Theorem 2**,

$$V_r V_r^T X^T X V_r V_r^T = V_* \Theta_r \Sigma_r^2 \Theta_r^T V_*^T$$
$$V_r \Sigma_r^2 V_r^T = (V_* \Theta_r) \Sigma_r^2 (V_* \Theta_r)^T$$
$$(V_r - V_* \Theta_r) \Sigma_r^2 (V_r - V_* \Theta_r)^T = 0$$
$$\|(V_r - V_* \Theta_r) \Sigma_r^2 (V_r - V_* \Theta_r)^T\|_F = 0$$

Using Frobenius norm to bound the matrix product,

$$\|\Sigma_r^2\|_F \|V_r - V_* \Theta_r\|_F^2 \geq 0$$

Since $\|\Sigma_r^2\|_F > 0$, equality is achieved if and only if $V_* \Theta_r = V_r$.

**Remark.** *Note that $\Theta_r$ is a rank-r unitary matrix wherein both rotation ($det(\Theta_r) = 1$) and reflection ($det(\Theta_r) = -1$) are valid since the order of the orthonormal vectors in the matrix $V_* \Theta_r = V_r$ do not alter $\|V_r - V_* \Theta_r\|_F$. In practice, $\Theta_r$ manifests itself predominantly as a rotation matrix during the iterative minimization using gradient descent.*

## C  ENERGY MINIMIZATION, LOSS SURFACE GEOMETRY, AND CONVERGENCE

In this section, we consider the energy minimization problem that constructs the projection space spanning the rank-$r$ sub-space of a given data matrix. For ease of visualization, we consider a $2 \times 2$ matrix $X = \text{diag}(5,1)$ with singular values 5 and 1 corresponding to right singular vectors $v_1 = [1,0]^T$ and $v_2 = [0,1]^T$, respectively. Our objective here is to extract a rank $r = 1$ approximation of this rank $f = 2$ matrix $X$. Certainly, this corresponds to identifying the right singular vector $v_1$ with singular value 5. The tail-energy surface (log-scale) corresponding to the $\|X\tilde{v}\tilde{v}^T - X\|_F$ is shown in **Fig. 10**. Here, $\tilde{v}$ is the test vector for a rank 1 approximation of X. The tail-energy is a bi-quadratic function in $\tilde{v}$ with 1 maximum, $2^r$ minima and $2^r$ saddle points, where $r$ is the desired low-rank approximation of a given data matrix. Furthermore, all minima have the same tail energy: a property of bi-quadratic functions. For the current specific example, the two equal tail-energy minima correspond to $v_1$ and $-v_1$, respectively.

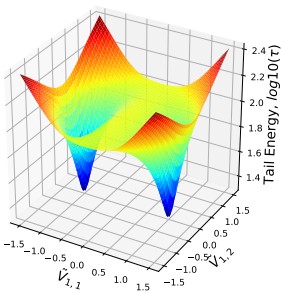

Figure 10: Surface plot for the bi-quadratic tail energy (log-scale) with 2 minima, 2 saddle points, and one maxima for $X = \text{diag}(5,1)$.

Although the minimization problem is non-convex, convergence is guaranteed since any perturbed-gradient descent approach converges to either of the stable fixed points (minima). In other words, any test vector $\tilde{v}$ other than $v_1$ or $-v_1$ will increase the tail-energy and hence will not be the solution. The same argument applies for a high-dimensional dataset $X$ where a low rank-$r$ approximation is desired with the number of equal tail-energy minima corresponding to $2^r$ for all possible negative and positive combinations of the $r$ right singular vectors $v_{i=1,\cdots,r}$. In effect, the stage 1 minimization problem constructs a right projection space $\tilde{V} = span\{v_1, \ldots, v_r\}$ that spans the top rank-$r$ subspace of a given dataset $X$. A similar line of argument then applies to our stage 2 minimization problem as well. A mild limitation, that will be addressed in our future work, occurs when $X = \text{diag}(5, 5 + \epsilon)$, $0 \leq \epsilon << 1$, wherein the two right singular values cannot be resolved accurately (still better than Randomized SVD methods) without further considerations. This latter case, with near algebraic multiplicity in singular values is a special case for conventional SVD as well.

## D  NETWORK INTERPRETABILITY

As described before in Fig. 2, our network weights and outputs are strictly defined and incorporated as losses in the network minimization problem. We therefore refer to the problem informed (SVD) restrictions on the network weights as representation driven losses. This is in contrast to kernel regularization loss often considered to impose a weak requirement on the network weights to be small. The representation driven, orthonormality loss term, in Stage 1 enforces that the weights $\tilde{V}$ must be orthonormal or $(V_*^T V_* = I_r)$ for a desired rank-$r$ which is greater than the rank-$f$ of the data matrix $X$. We numerically verify the interpretability of the layer outputs and weights by considering two networks: (1) with, and (2) without the aforementioned orthonomality loss. For each of these two cases, two synthetic datasets are considered corresponding to $r \leq f$ and $r > f$. Please note that in a practical scenario $f$ is an unknown and can be determined only by performing a full SVD of $X$. Therefore, numerically testing this aspect for our solver is necessary.

For the first case, we consider a synthetic data matrix $X_{15 \times 15}$ where the top 5 singular values are positive ($f = 5$) while the rest are zero. The objective is to extract the top 10 ($r = 10$) singular vectors where the desired rank is higher the the rank of the system itself. A total of four training runs

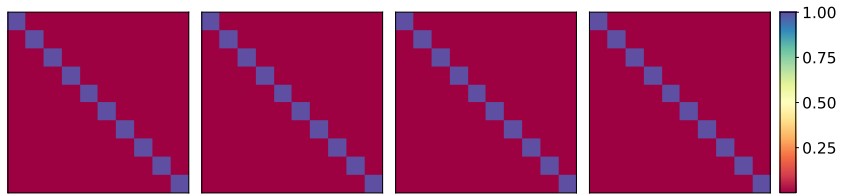

Figure 11: Synthetic Low Rank Case: Correlation Map of extracted vectors $V_*$ over four runs. Orthonormality is imposed for the first run resulting in a diagonal structure. Absence of this condition results in a scatter.

are considered: one run for a network with the orthonormality condition imposed and three runs for another network without this additional constraints. **Fig. 11** shows the correlation map between the recovered vectors $V_*$ for each of the four runs. Notice that only when the orthonormality criteria is not imposed, we get scatter away from the diagonal matrix, although all four runs converged to the same tail energy. Since the true rank of $X$ is 5, the null space of $X$ is of dimension 10. The absence of this orthonormality imposing, representation loss results in non-orthonormal vectors $V_*$ spanning the low-rank range space.

Figure 12: Synthetic Full Rank Case: Correlation Map of extracted vectors $\tilde{V}$ over four runs. Orthonormality loss now does not contribute and therefore all runs have a diagonal structure.

For the second case, we consider a full-rank, synthetic data matrix $X_{15 \times 15}$ with $f = 15$. As before, we extract the top 10 singular vectors ($r = 10$) using four training runs: one with and three without imposing the orthonormality loss. Since the desired rank 10 system now itself is full rank, this additional loss term does not contribute, as desired. **Fig. 12** shows that the extracted vectors $V_*$ remain orthonormal, for all the four runs, so as to minimize tail energy ($\|X(\tilde{V}\tilde{V}^T - I)\|_F$), as described in **Section 3.1** above. In fact, for any rank $r$ approximation of a rank $f$ system such that $r \leq f$, an arbitrary non-orthonormal matrix $V_*$ will increase the tail energy and hence will not be a fixed point (solution) of our network minimization problem.

## E    IMPLEMENTATION DETAILS

### E.1    CHOICE OF ACTIVATION FUNCTIONS

All the activation functions in both stages are *Linear* with no biases. One might argue that this choice is not a neural approach, since all the activations are linear. However, please note that singular vectors are *linearly separable orthogonal features* of a dataset, and therefore any other choice of activation function will result in approximation errors. Since the elements of singular vectors are in $[-1, 1]$, a choice of *relu* activation is problematic. A simple verification is to approximate a straight line segment in $[-1, 1]$ with *tanh* activation, only to realize that the approximation error $\to 0$ as the number of neurons $\to \infty$. These arguments can also be verified by replacing linear activation in Stage 1 by any non-linear activation only to find that the tail energy bound cannot be satisfied. Note that given a small matrix, one can calculate the right singular vectors and substitute them directly as our network weights to confirm this tail energy bound.

### E.2    TRAINING, VALIDATION, AND TESTING SPLIT

An issue with training and validation split in matrix decomposition problems is that the error norm cannot be bounded in a deterministic manner or computationally verified. For Singular Value Decomposition of a given data matrix $X$ differs from SVD on a truncated dataset $\hat{X}$ in it's singular triplets (singular values and vectors). Ensuring these triplets do not change over an arbitrary split is a non-trivial computational task.

**Remark.** *For dataset $X$, an arbitrary training/validation split results in a varying dataset $\hat{X}$ wherein the norm $\|X_{pred} - \hat{X}\|_F$ changes according to the split. Since the desired features are unknown a priori, a consistent truncated dataset $\hat{X}_c$ that spans the same space as the full data $X$ cannot be obtained using an arbitrary split.*

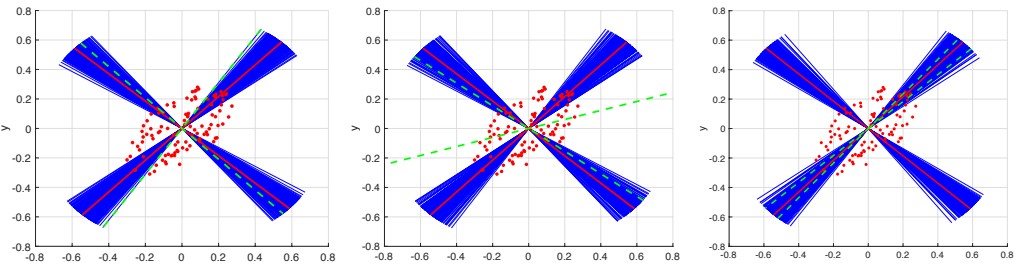

Figure 13: Variation in extracted right singular vectors (dashed green lines) for three runs with each run comprising of 200 different train/test data splits with an $80/20$ ratio. The solid red lines indicate the ground truth singular vectors when using the entire unsplit dataset (red dots). Extracted singular vectors deviate increasingly from left to right over the three runs.

This results in a large variance in extracted features over multiple training/testing splits since the span of $\hat{X}$ itself is changing with each split. **Fig. 13** shows a numerical experiment with a synthetic (rank-2) dataset containing 100 samples drawn (red dots) from a 2D ellipse with major and minor axes 2 and 1, respectively. The ground truth right singular vectors over the unsplit dataset are computed using conventional SVD (solid red lines). Next the dataset was split with an $80/20$ ratio as the training/testing split for 200 different realizations per run. The mean/expectation of the extracted right singular vectors (dashed green line) over the 200 realizations are then reported. One can easily see that the expected right singular vectors vary quite substantially compared to the ground truth over the three runs indicating that an arbitrary split of the dataset does not ensure accuracy.

### E.3   DATA STREAMING

Given a data matrix $X^{m \times n}$, we stream the data along the smaller dimension assuming the user prescribed rank-$r$ is such that $r \leq min(m, n)$. For the sake of simplicity we assume that the data matrix has $m$ samples and $n$ features, where $m > n$, and consequently feature vectors of samples are streamed in batches. We rely upon the built-in Keras **fit_generator** class for data streaming from the secondary memory (HDD). For a big data matrix $X$ that cannot be loaded into the main memory, this allows us to mimic the modality of data residing on an external server. Thus, given a pointer to the data, the function yields a batch of specified size for the network to train on for specific epochs. This ability saves main memory load and allows us to process bigger datasets on smaller main-memory machines than reported in prior works.

Note that for the stage-1 network to converge to a desired tolerance, we require multiple passes (empirically $\leq 5$) over the original data streamed batchwise. Therefore for Stage 1, the input data is never persistently present in the main memory of the remote machine. The output data is dumped onto the secondary memory assuming that storing a low rank approximation is still main memory intensive. For Stage 2, this low rank approximation in the secondary memory is streamed as input, and the extracted singular values and vectors are saved in main memory.

### E.4   SETUP AND TRAINING

All experiments were done on a setup with Nvidia 2060 RTX Super 8GB GPU, Intel Core i7-9700F 3.0GHz 8-core CPU and 16GB DDR4 memory. We use Keras (Chollet, 2015) running on a Tensorflow 2.0 backend with Python 3.7 to train the networks presented in this paper. For optimization, we use AdaMax (Kingma & Ba, 2014) with parameters (*lr*= 0.001) and 1000 steps per epoch.

### E.5   ERROR METRICS

As discussed before in **Section 2.1**, since relative errors in tail energies do not imply similar errors at scales in extracted singular factors, we rely upon additional error metrics on the extractor factors for

performance comparison and benchmarking. In the following $X$ and $\hat{X}$ are used to denote the true and the reconstructed data matrices.

- **Scree Error:** Absolute difference between true and approximated singular values.

$$scree_{err}(r) = |\sigma_i(X) - \hat{\sigma}_i(\hat{X})| \quad \forall\, i \in [1, r]$$

- **Reconstruction Error:** Frobenius norm error of the true data and its rank-$r$ approximation.

$$frob_{err}(r) = \|X - \hat{X}\|_F^2 - \|X - X_r\|_F^2$$

- **Spectral Error**: 2-norm of the singular value of the true data and its rank-$r$ approximation.

$$spectral_{err}(r) = \|X - \hat{X}\|_2 - \|X - X_r\|_2$$

- **Chi Square Statistic**: Deviation between the true and approximated singular vectors.

$$\chi_e^2 rr(r) = 1 - \frac{1}{r}\|Corr(v_{[:r]}(X), \hat{v}_{[:r]}(\hat{X}))\|_F$$

Here, $\sigma_i$s are the true singular values and $X_r$ is the desired rank-$r$ approximation of $X$ using conventional SVD as the baseline for benchmarking. Under perfect recovery, all error metrics are expected to approximately achieve zero at machine precision. All of our numerical experiments were performed on a GPU using single (32-bit) precision floating point operations. Therefore, the tail energies are expected to be correct to up to 8 significant digits in all the subsequent calculations. In the following sections, $\hat{X}$ is replaced by approximations from SketchySVD and Range-Net.

### E.6 Loss Profile

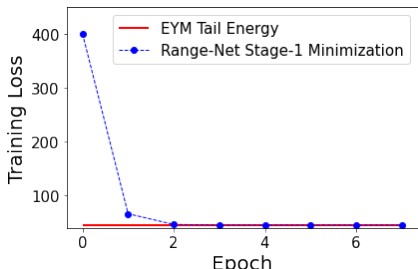

Figure 14: Range-Net Stage-1 training loss for Parrot image. The network minimization loss converges to the EYM Tail Energy bound within five epochs and stabilizes for further epochs.

## F Additional Experiments

### F.1 Feature Extraction: Graph (Eigen)

Table 5: Description and Metrics of the Network Graphs for Range-Net

| Dataset | Nodes | Edges | rank | $err_{fr}$ | $err_{sp}$ | $\chi_{err}^2$ |
|---|---|---|---|---|---|---|
| Airlines (air) | 235 | 2101 | 200 | 0 | 0 | 0.011 |
| Twitter (air) | 3556 | 188712 | 200 | 0 | 0 | 0.014 |
| Wikivote (Leskovec & Krevl, 2014) | 8297 | 103689 | 200 | 0 | 0 | 0.027 |
| Wikipedia (lev) | 49728 | 941425 | 100 | 4.27e-6 | 1.23e-7 | 0.034 |
| Slashdot (Leskovec & Krevl, 2014) | 82168 | 948464 | 100 | 8.56e-6 | 6.92e-7 | 0.045 |

Large scale networks occur in many applications where SVD is primarily used to identify the most important nodes or as a pre-processing step for community detection. For these kind of graph based datasets, we either perform SVD or Eigen decomposition on the graph, depending on the format in which the data arrives. We demonstrate results on the following graphs of varying size, tabulated in **Table 5**. If the data arrives directly in the form of an adjacency matrix, we can perform SVD or Eigen decomposition on it directly. For cases, where an adjacency list is provided, a pre-processing step is required to convert the list representation in a sparse vector.

Since an Eigen decomposition problem is a special case of SVD, where the data matrix is symmetric positive semi-definite, Range-Net is directly applicable. The benchmark was generated for smaller graphs using a conventional SVD solver. For larger graphs, a similar benchmark was constructed using the *irlba* routine by Baglama & Reichel (2005). **Table. 5** shows the error metrics for all the graphs, where consistently low values are observed for Frobenius and Spectral error metrics.

### F.2 NAVIER-STOKES SIMULATED DATA (SVD)

For our next numerical experiment, we rely upon synthetic data generated using a Navier Stokes flow simulator for an incompressible fluid. The flow data is available on tensor-product grid on two-spatial and one temporal dimensions of size $(w, h, t) = (100 \times 50 \times 200)$. For each point on the grid, velocity vector values are available in both $x$ and $y$ spatial dimensions for 200 time instances. The fully-developed, flow pattern exhibits a periodicity in the time dimension at approximately every $\sim 60$ time step that can be identified using SVD as characteristic modes. The data is therefore reshaped into a spatial vector for each time instance resulting in a spatio-temporal matrix $X \in \mathbb{R}^{5000 \times 200}$. For comparison purposes, we use only the x-direction stream velocity.

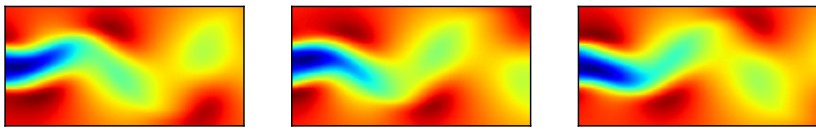

Figure 15: From left to right: x-direction stream velocity at times t = 0, 100, and 200

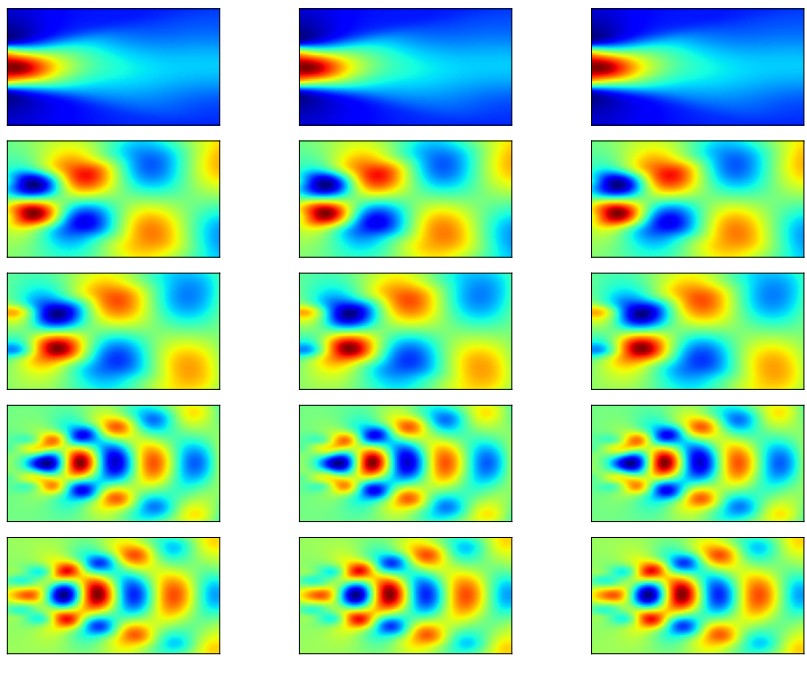

(a) SketchySVD          (b) Truth          (c) Range-Net

Figure 16: Reshaped $U$ indicative of dynamic modes, corresponding to top six right singular vectors for $r = 5$.

**Fig. 15** shows the evolution of the stream velocity over three time instances and the inherent time-periodic nature of the data. Notice the central-left region of primary flow across all three images. **Fig. 16** shows the reshaped $U$ vectors for SketchySVD, conventional SVD and Range-Net. Notice that the images corresponding to the first left singular mode (also called dynamic modes) captures a notion of the primary flow in the left-center part. The second one captures spatial variations of the flow as time progresses. For all the three methods, all the modes have similar solution visually. Note that

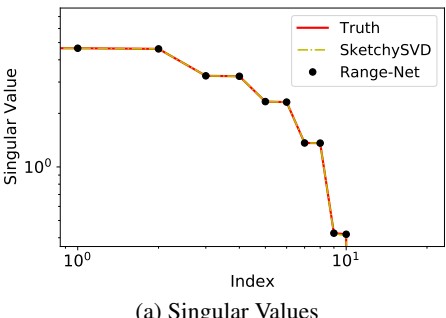 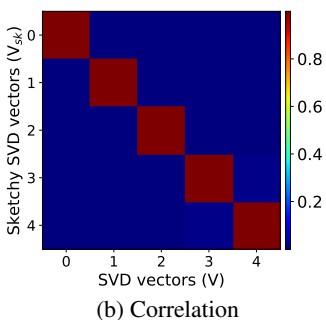

(a) Singular Values  (b) Correlation

Figure 17: SketchySVD and Range-Net (a) extracted singular values and (b) cross correlation between estimated and true (conventional SVD) right singular vectors for $r = 5$ on the Navier-Stokes data.

Range-Net computes a rank $r = 5$ approximation without oversampling, wherein SketchySVD relies upon memory intensive sketchy projections of ranks $k = 21, s = 43$ to arrive at the solution. Overall, for this low-rank dataset both SketchySVD and Range-Net perform reasonably well qualitatively looking at the features in **Fig. 16** and the singular value spectrum in **Fig. 17**, because this synthetic dataset is extemely low-rank ($f = 10$).

**Remark.** *For a given data matrix $X \in \mathbb{R}^{m \times n}$ of rank-$f$ ($f \leq \min(m, n)$) SketchySVD generates low error approximations if the oversampled rank $k$ is such that $k \geq f$. For full rank tall skinny matrices, this implies that $k \geq \min(m, n)$. For cases where $k < f$ (full rank or otherwise), SketchySVD accrues large approximation errors resulting in incorrect SVD factors.*

From a use-case point of view, randomized SVD generates low-error factors for full-rank, tall skinny matrices ($X \in \mathbb{R}^{m \times n}$) only when the oversampled rank $k \geq n$. This poses a serious limitation for all applications where this requirement is not met and consequently randomized SVD algorithms accrue large approximation errors in SVD factors as shown in the Sandy Big Data case study below and **Appendix F.4**. Note that given a big data matrix $X$ determining the rank $f$ of $X$ is unknown and therefore selecting an oversampled rank $k$ such that $k \geq f$ is impractical in such cases.

**Remark.** *Once the SVD factors are extracted, SketchySVD cannot be independently verified without performing a full SVD. In contrast, Range-Net is independently verifiable since Stage-1 of Range-Net cannot return orthonormal vectors if the vectors do not span the rank-$r$ subspace of a given data $X$.*

### F.3   HIGH RANK APPROXIMATION: SANDY BIG DATA

This section provides an addendum to the numerical results presented in **Section 4.3** of the main text. **Fig. 18** shows evolution of Hurricane Sandy for two time instances. Similar to the Navier-Stokes simulation data, **Fig. 19** shows three dynamic modes corresponding to rank $1, 20, 50, 100$ singular values. As shown, our results are in good agreement with conventional SVD whereas, Sketchy SVD shows substantial deviations after the first 50 dynamic modes. **Fig. 20** shows the scree-error in the singular values extracted by SketchySVD and Range-Net.

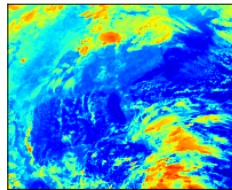 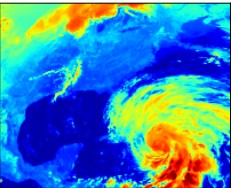

Figure 18: Satellite image captures of hurricane Sandy over the Atlantic ocean at $t = 0$ (left) and $t = 200$ minutes approximately (right).

**Fig. 21** shows the cross correlation of the right singular vectors and scree-error in the corresponding singular values extracted by SketchySVD and Range-Net. Note that for a rank-100 approximation, SketchySVD extracted right singular vectors start deviating after rank-10 as shown in **Figs. 21** while the singular values deviate quite substantially from rank-1. Range-Net on the other hand is in excellent agreement with the right singular vectors and values for all desired 100 indices.

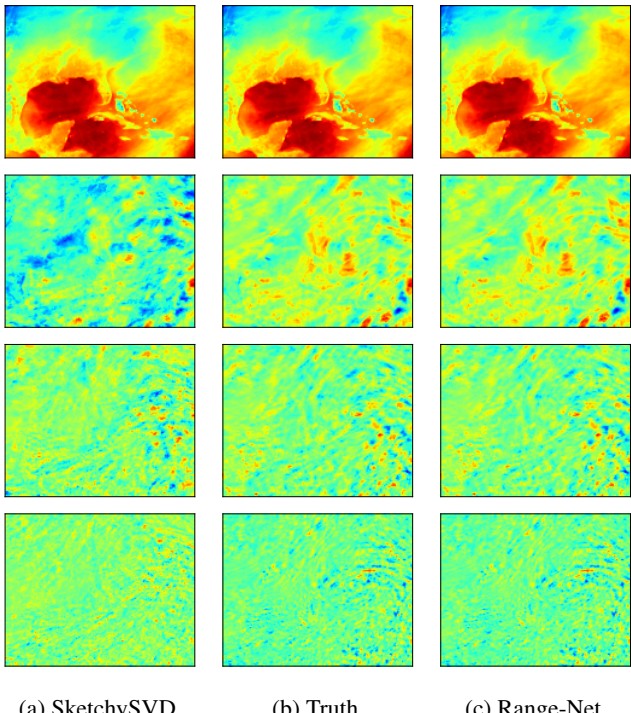

|                |            |              |
| :------------: | :--------: | :----------: |
| (a) SketchySVD | (b) Truth  | (c) Range-Net |

Figure 19: Reshaped $U_i$ indicative of dynamic modes, corresponding to $i = 1, 20, 50, 100$ for $r = 100$ (oversampled rank $k = 401$ for SketchySVD. The dynamic mode approximation error stand out visually for SketchySVD for indices $20, 50, 100$. Our method does not have such artifacts.

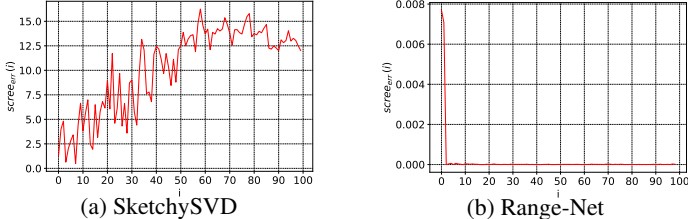

|                |              |
| :------------: | :----------: |
| (a) SketchySVD | (b) Range-Net |

Figure 20: Scree-error in singular values for (a) SketchySVD and (b) Range-Net where a conventional SVD is used as the baseline in scree-error metric.

We point out that accuracy is a matter of special concern in scientific computations. Any compression that results in substantial loss of information or obscuring an otherwise identifiable feature in recorded observations directly culls our capacity to make scientific improvements. Consequently, any exploratory data analysis, however big or small, must accurately identify the underlying features. Range-Net achieves the lower bound on tail-energy given by EYM theorem to ensure an accurate resolution in big data setting. Note that increasing the sensor resolution implies that we are interested in exploring and understanding the high-frequency features (lower singular values) of the data.

### F.4   LOW RANK APPROXIMATION: SANDY BIG DATA

In this experiment, we extract the rank-10 SVD factors using SketchySVD and Range-Net for the Sandy dataset. The oversampled ranks for sketchy SVD are $k = 4r + 1 = 41$ and $s = 2k + 1 = 83$ where $k, s \ll \min(m, n)$.

As before, **Fig. 22** show the cross-correlation between the extracted and true (conventional SVD) right singular vectors using SketchySVD and Range-Net. **Fig. 23** shows the scree error in the extracted singular values for the two methods with singular values from conventional SVD as the baseline. Finally, **Fig. 24** shows a comparison between extracted dynamic modes corresponding to indices $i = [1, 4, 7, 10]$ from SketchySVD, conventional SVD, and Range-Net. One can easily see

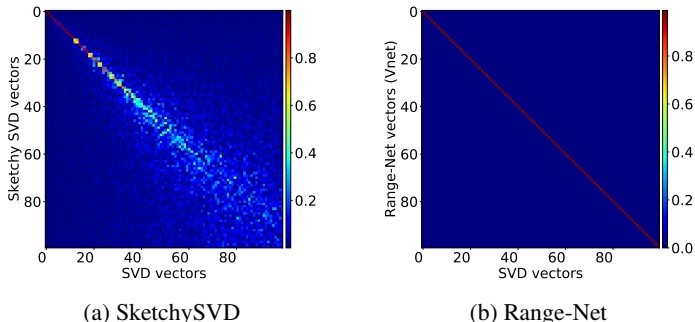

(a) SketchySVD            (b) Range-Net

Figure 21: Cross-correlation between extracted and true (conventional SVD) right singular vectors for (a) SketchySVD and (b) Range-Net for a rank $r = 100$ approximation. SketchySVD deviates substantially after index 10 (although sketching at sizes $k = 401$ and $s = 803$) while Range-Net is in good agreement for all the 100 indices.

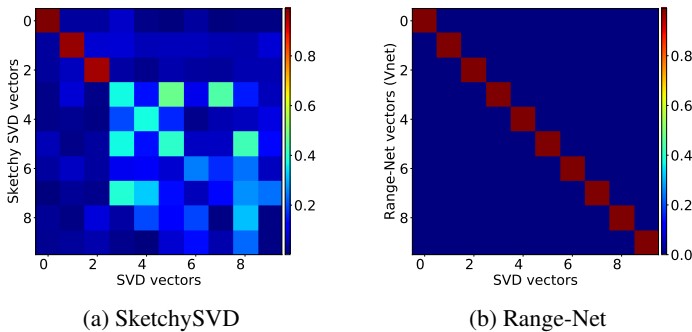

(a) SketchySVD            (b) Range-Net

Figure 22: Cross-correlation between extracted and true (conventional SVD) right singular vectors for (a) SketchySVD and (b) Range-Net for a rank $r = 10$ approximation. SketchySVD deviates substantially after index 3 while Range-Net is in good agreement for all the 10 indices.

that Sketchy SVD extracted dynamic modes/right singular vectors deviate quite substantially for $i = [4, 7, 10]$.

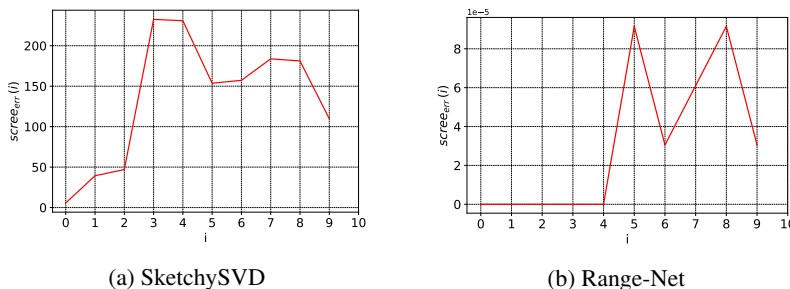

(a) SketchySVD            (b) Range-Net

Figure 23: Scree-error in singular values for (a) SketchySVD and (b) Range-Net where a conventional SVD is used as the baseline in scree-error metric. Note that for Range-Net the error is at a scale of $10^{-5}$, 7 orders of magnitude apart from SketchySVD ($10^2$).

## F.5 PARROT: ADDENDUM CORRELATION OF LEFT SINGULAR VECTORS

In the following, we show the deviation of left singular vectors, computed using Range-Net and Sketchy SVD, from the conventional SVD computed left singular vectors as a baseline. As shown before, random sketching introduces irreducible errors in randomized SVD methods resulting in this unchecked deviation. **Fig. 25** shows a comparison of the cross-correlation against the common baseline for both Range-Net and Sketchy SVD.

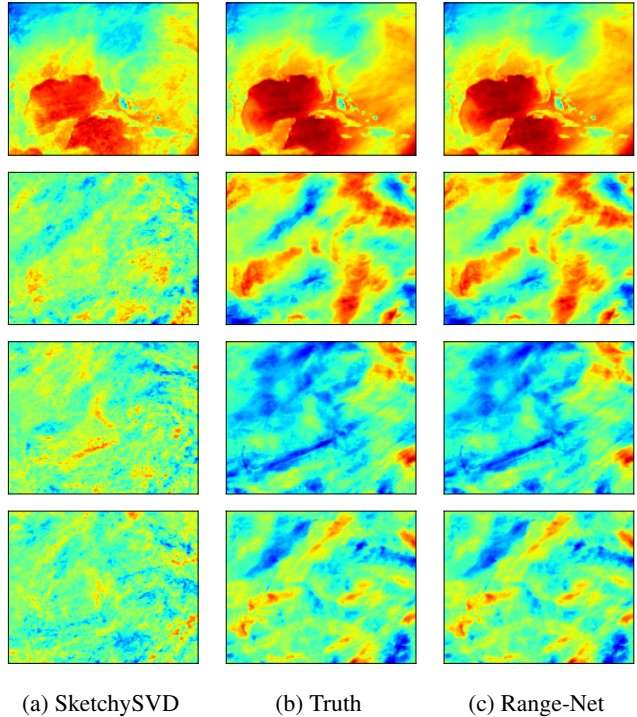

(a) SketchySVD          (b) Truth          (c) Range-Net

Figure 24: Reshaped $U_i$ indicative of dynamic modes, corresponding to $i = 1, 4, 7, 10$ for $r = 10$. The dynamic mode approximation error stand out visually for SketchySVD for indices $1, 4, 7, 10$. Our method does not have such artifacts.

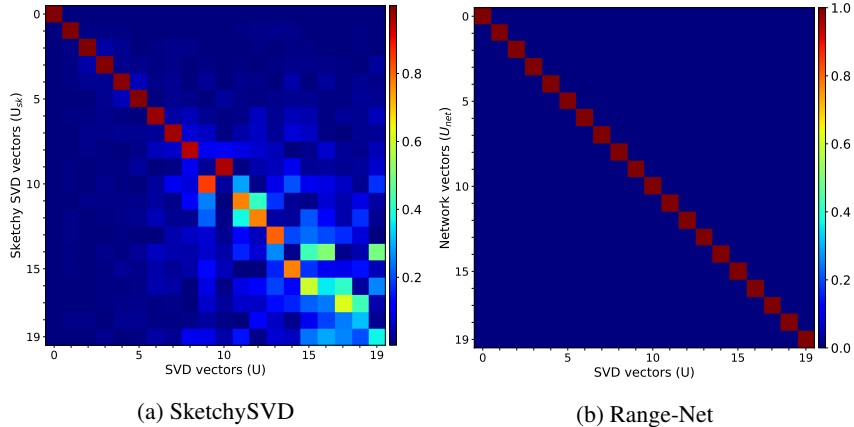

(a) SketchySVD          (b) Range-Net

Figure 25: Cross-correlation between true (conventional SVD) and extracted left singular vectors from (a) SketchySVD (b) Range-Net for a rank-$r = 20$ approximation of the Parrot image.

# G SKETCHY SVD IMPLEMENTATION

A brief outline of the single-pass Sketch-SVD algorithm from Tropp et al. (2019). Note that for the numbers reported in terms of storage, we implemented this code with additional memory optimization and sparse matrices.

---

**Algorithm 1** Sketchy SVD

---

**Input:** $X \in \mathbb{R}^{m \times n}, r$ : expected rank
**Output:** $\tilde{X} \in \mathbb{R}^{m \times k}$ the approximated rank $k$-dim data

1: Initialize $k = 4r + 1, s = 2k + 1$                          ▷ Oversampling parameters
2: Projection maps: $\Upsilon \in \mathbb{R}^{k \times m}, \Omega \in \mathbb{R}^{k \times n}, \Phi \in \mathbb{R}^{s \times m}, \Psi \in \mathbb{R}^{s \times n}$
3: Projection matrices: $A \in \mathbb{R}^{k \times n}, B \in \mathbb{R}^{m \times k}, Z \in \mathbb{R}^{s \times s}$ as empty
4: **for** $i = 1 : n$ **do**                                    ▷ Streaming Update
5:      Form $H \in \mathbb{R}^{m \times n}$ as a sparse empty matrix
6:      $H(i, :) = X(i, :)$                                       ▷ Streamed columns
7:      $A \leftarrow A + \Upsilon H$                             ▷ Update Co-Range
8:      $B \leftarrow B + H\Omega^T$                              ▷ Update Range
9:      $Z \leftarrow Z + \Phi H \Psi^T$                          ▷ Update Core Sketch
10: $Q \in \mathbb{R}^{m \times k} \leftarrow qr\_econ(B)$        ▷ Basis for Range
11: $P \in \mathbb{R}^{n \times k} \leftarrow qr\_econ(A^T)$      ▷ Basis for Co-Range
12: $C \in \mathbb{R}^{s \times s} \leftarrow ((\Phi Q) \setminus Z)/(\Psi P)$   ▷ Core Matrix
13: $[U, \Sigma, V^T] \leftarrow svd(C)$                          ▷ Full SVD of Core Matrix
14: $\Sigma \in \mathbb{R}^{r \times r} \leftarrow \Sigma[1 : r, 1 : r]$   ▷ Pick top $r$
15: $U \in \mathbb{R}^{k \times r} \leftarrow U[:, 1 : r]$        ▷ Pick top $r$
16: $V^T \in \mathbb{R}^{r \times k} \leftarrow V^T[1 : r, :]$    ▷ Pick top $r$
17: $U \in \mathbb{R}^{m \times r} \leftarrow QU$                 ▷ Project to Row Space
18: $V^T \in \mathbb{R}^{r \times n} \leftarrow PV^T$             ▷ Project to Column Space
19: $\tilde{X} \in \mathbb{R}^{m \times n} \leftarrow U\Sigma V^T$   ▷ Approximation

---

