# OpenReview forum: "Range-Net: A High Precision Neural SVD"
_ICLR.cc/2022/Conference — ICLR 2022 Submitted_

### Official Review · Reviewer_YsEC · 2021-10-20

**Correctness:** 3
**Technical Novelty And Significance:** 2
**Empirical Novelty And Significance:** 2
**Recommendation:** 5
**Confidence:** 4

**Main Review:**

##########################################################################

Summary: This paper offers an interesting approach to scaling rank r approximation of a matrix.

##########################################################################

Reasons for score: I am inclined to voting for a reject (borderline). The proposed algorithm is intuitive and simple - but I think the experimental sections could be augmented with more results. I am open to changing the rating, pending the authors' responses.

##########################################################################

Pros:
1. Proposes an interesting algorithm for solving a problem widely applicable in machine learning.


##########################################################################

Cons:
1. It seems the trained neural network is specifically tied to a particular dataset. Can a trained neural network be reused for another dataset of the same size? (e.g. time series data)

2. There are some lingering questions on experiments. Why does range-net require more memory than sketchy SVD on dimension reduction (Table 3)? How much is the deviation in the computed singular vectors from orthogonality?

3. How is the neural network trained? I did not see much description in the submission.


##########################################################################

Questions during rebuttal period:

Please provide clarifications for the points in the previous section. I also did not see an appendix in the submission which I wanted to glance at.

#########################################################################

**Summary Of The Paper:**

This paper proposes a neural network architecture for computing a rank-r approximation of a matrix. The first part consisted of learning the projection and the second part consisted of learning the rotation. An experimental comparison against the state-of-the-art Sketchy SVD algorithm is provided on three datasets. The authors claim that the proposed method is extremely memory efficient compared to previous approaches.

**Summary Of The Review:**

While the paper proposes an interesting algorithm, in the end I did feel much void in the experimental section. Perhaps adding more dimensions in terms of algorithm comparison (one more algorithm in addition to sketch SVD) / adding more datasets could help. In addition, the authors could clarify how trained neural network can be adapted to not just one dataset but a dataset that may be related (time series).

---

> ### Author Response · Authors · 2021-11-17
> **Comments for Reviewer YsEC**
>
> 1. Pros: Proposes an interesting algorithm for solving a problem widely applicable in machine learning.
>
> The appendix was provided as a supplemental material due to the page limit restriction and desk reject policy of ICLR. Please let us know if you are able to access it. We have also added a easy to use Python code base with implementation of a number of randomized methods (a few suggested by other reviewers in addition to the ones presented in the main text) to numerically demonstrate the errors.
>
> 2. It seems the trained neural network is specifically tied to a particular dataset. Can a trained neural network be reused for another dataset of the same size? (e.g. time series data)
>
> SVD on a data matrix $X$ is only for the dataset itself. For example, consider a trivial matrix $X = \begin{bmatrix}
> 1 & 0;\\
> 0 & 1
> \end{bmatrix}$ where the singular vectors form an identity matrix with two equal singular values 1. The vectors or values from this cannot be used to predict the singular vectors or values of another matrix $X = \begin{bmatrix}
> 1 & 1;\\
> 0 & 1
> \end{bmatrix}$ which has distinctly different vectors and values. The only similarity is that both have a rank-2 other than that nothing can be used from one for the other.
>
> 3. There are some lingering questions on experiments. Why does range-net require more memory than sketchy SVD on dimension reduction (Table 3)? How much is the deviation in the computed singular vectors from orthogonality?
>
> I think there might be slight confusion here. Our memory consumption in Table 3 and Table 4 of the main text is in megabytes (MB) whereas SketchySVD is in gigabytes (GB).
>
> Please refer to Appendix D for details regarding the deviation from orthogonality of the extracted singular vectors. Since we baseline against conventional full SVD solver to verify our results, a cross-correlation map is shown for Range-Net and conventional SVD extracted right singular vectors. In fact, our numerical verification relies upon orthogonality (identity) as per Theorem 2 to confirm our results.
>
> Please also refer to the correlation map shown in Fig. 6 (a,b) (Additional figures for other datasets in Appendix Fig. 21, 22) in main text where we have plotted the deviation from orthogonality of the right singular vectors of SketchySVD and Range-Net against the baseline (conventional, full SVD) right singular vectors for a fair comparison.
>
> 4. How is the neural network trained? I did not see much description in the submission.
>
> The details are presented in the Appendix in the supplementary material. Please also take a brief look at the additional supplemental code now provided that provides further clarity in terms of errors accrued by randomized SVD methods.
>
> 5. Please provide clarifications for the points in the previous section. I also did not see an appendix in the submission which I wanted to glance at.
>
> Please let us know if you are able to access the appendices in the supplementary material. We added it separately before and have updated it once again for further clarity.
>
> 6. While the paper proposes an interesting algorithm, in the end I did feel much void in the experimental section. Perhaps adding more dimensions in terms of algorithm comparison (one more algorithm in addition to sketch SVD) / adding more datasets could help. In addition, the authors could clarify how trained neural network can be adapted to not just one dataset but a dataset that may be related (time series)
>
> Since the page limit is strict, a number of expanded experimental results were provided in the Appendix along with the accompanying theoretical results. The appendix also contains training details on hurricane Sandy dataset which is a time-evolving image data. SVD is used here to perform a time domain decomposition where the dominant modes correspond to the higher singular values.
>
> The supplemental material also contains additional datasets including graphs and flow data to showcase the accuracy of Range-Net in terms of tail energy and extracted SVD factors.

---

### Official Review · Reviewer_MNXm · 2021-10-30

**Correctness:** 3
**Technical Novelty And Significance:** 2
**Empirical Novelty And Significance:** 2
**Recommendation:** 3
**Confidence:** 5

**Main Review:**

Strengths:
	RangeNet just costs O(nr+r^2) space to calculate truncated SVD, which is memory optimal. Besides, RangeNet produces the nearly accurate results with no more than 5 passes on matrix, which is pass efficient.

Weakness:
1. No time analysis of real runtime is listed in this paper. Once this algorithm costs much time to produce results, it is not feasible in real applications.
2. There is no dataset shown which can be handled by RangeNet but cannot be handled by classical algorithms for truncated SVD. Experiments on large dataset will make this work more solid.
3. It is not fair to compare RangeNet with SketchSVD, RangeNet just produces the right singular vectors while SketchSVD produces both left and right singular vectors. Besides, the experiments in paper just compare them on matrix with size m >> n.

Some recent work on randomized truncated SVD should be compared, like:

Efficient randomized algorithms for the fixed-precision low-rank matrix approximation
W Yu, Y Gu, Y Li - SIAM Journal on Matrix Analysis and Applications, 2018 - SIAM

Fast randomized PCA for sparse data
X Feng, Y Xie, M Song, W Yu… - Asian conference on …, 2018 - proceedings.mlr.press

**Summary Of The Paper:**

In this paper Range-Net is proposed for calculating the right singular vectors and singular values, which produces nrealy accurate results with less memory cost than existing SkectchSVD. However, no time analysis or real runtime is listed in this paper, which is an important shortcoming of this paper.

**Summary Of The Review:**

Without runtime listed in this paper, although this algorithm is memory optimal, it is not good enough. Besides, using linear optimizer to solve the truncated SVD seems not a novel idea.

---

> ### Author Response · Authors · 2021-11-17
> **Comments for Reviewer MNXm**
>
> 1. RangeNet just costs $O(nr+r^2)$ space to calculate truncated SVD...
>
> We are thankful for the reviewer's comments.
>
> 2. No time analysis of real runtime is listed in this paper... it is not feasible in real applications.
>
> We agree with the reviewer and have added runtimes for our approach. Please note that a run-time comparison between two approaches makes sense only when the two approaches generate equally accurate results. In other words, if the end-user can specify the error tolerance for the two methods then the run-times are a direct indication of which method is better. However, our theoretical results (Theorem 2) and numerical experiments show that randomized-SVD errors (for a given rank-r) cannot be reduced making a direct comparison difficult. As requested, please find the runtime for Range-Net and SketchySVD in Tables 2, 3 and 4.
>
> 3. There is no dataset shown which can be handled by RangeNet... Experiments on large dataset will make this work more solid.
>
> We rely on hurricane Sandy as a big-data example (24 GB). The numerical experiment is performed on a truncated Sandy dataset since a baseline comparison using conventional full SVD can only be performed on our machine (16 GB RAM) where the peak RAM load of conventional SVD is near 16GB. The reviewer is referred to Section F.3, Fig. 19, 20, and 21. Certainly, Range-Net can handle larger datasets but the error comparison against the well known full SVD is not feasible.
>
> 4. It is not fair to compare RangeNet with SketchSVD, RangeNet just produces the right singular vectors while SketchSVD produces both left and right singular vectors. Besides, the experiments in paper just compare them on matrix with size m >> n.
>
> Range-Net computes both left and right singular vectors but does not consume main memory to store left singular vectors at run-time. Please refer to Fig. 25 that shows the deviation in the left singular vectors $U$ for the Parrot case in Appendix F.5. For Randomized SVD methods, the deviation in $U$, please refer to the plots in the Jupyter notebook provided in Supplemental material.
>
> Note that for a rank-r approximation with right singular vectors $V_{r}$ and values $\Sigma_{r}$, $U_{r}= XV_{r}(\Sigma_{r})^{-1}$. This has been described in detail in \textbf{Section 3.1, page 6}. For fair comparison, we also show that SketchySVD works well for tall and skinny matrices $(m\gg n)$. Please see the numerical experiment in Appendix F.2 where we show this.
>
> For Parrot image (Section 4.1), we have $m=1024$ and $n=1536$. Similarly, for the synthetic diagonal case (Appendix A), we have $m = n =500$. We have shown numerical experiments for both $m\approx n$ and $m \gg n$. Please refer to Section 4.4 (Storage Complexity Analysis) where $m=50k$ is held while $n$ ranges from $10k$ to $150k$.
>
> 5. Some recent work on randomized truncated SVD should be compared, like: Efficient randomized algorithms for the fixed-precision low-rank matrix approximation W Yu, Y Gu, Y Li - SIAM Journal on Matrix Analysis and Applications, 2018 - SIAM
> Fast randomized PCA for sparse data X Feng, Y Xie, M Song, W Yu… - Asian conference on …, 2018 - proceedings.mlr.press
>
> We have provided Python code (see supplementary material notebook) for a number of randomized SVD algorithms along with errors in their respective tail-energies and SVD factors to further ratify our statements. Please note that this code base contains implementations of ``Fast randomized PCA for sparse data" (frPCA) in addition to the existing algorithms we compared against:  SketchySVD, BlockLanczos, RandSVD.
>
> Also please note that frPCA explicitly computes the covariance matrix of the data as $X^TX$, which simply does not work for a big data setting, when even the full data might not be loaded into memory. In comparison, please refer to the MNIST Eigen (Section 4.2), where Range-Net also computes the eigen factors, without ever constructing the covariance matrix.
>
> However, the first citation suggested by the reviewer "Efficient randomized algorithms for the fixed-precision low-rank matrix approximation" is a randomzied QR type decomposition, hence not comparable with our problem setting i.e., SVD.
>
> As shown in Theorem 2, any projection of the original data matrix onto a sketching matrix that is different from $V_{r}V_{r}^{T}$, where $V_{r}$ is the top rank-$r$ right singular vector matrix will always result in errors, both in tail energy and SVD factors. The provided code numerically ratifies this theoretical result. We would also like the reviewer to see the remark made in Page 4 of main text, wherein (Musco and Musco 2015 NIPS) states that a low relative error in tail energy does not imply the extracted singular values and vectors will have similar relative errors at scale. Additionally, we are advocating for reporting of absolute tail errors instead of relative tail errors, because Frobenius errors of practical data matrices are large, hence a visual check can be erroneous.

---

### Official Review · Reviewer_vVFi · 2021-11-06

**Correctness:** 4
**Technical Novelty And Significance:** 3
**Empirical Novelty And Significance:** 3
**Recommendation:** 6
**Confidence:** 3

**Main Review:**

- The extent to which the choice of the hyper-parameters of AdaMax (l_r and #steps per epoch) affects the result should be made more clear. Currently, the authors simply state those choices in the Appendix without justifying them. Also, the robustness of the underlying methodology w.r.t. the exact choice of optimization framework is also not clear (e.g., AdaMax or some other variant).

- There is very limited discussion motivating the target use cases; it'd be helpful if the authors could add more detail in terms of ensuring the usefulness of having those low-rank approximations computed is clear to the reader (e.g., Sandy Big Data use case).

- The authors could add more detail on why only SketchySVD was used as a baseline; also I believe the following work is a fundamental one in the area, so might worth citing that for the reader to get a more complete view of the topic [not my paper]:
Liberty, Edo. "Simple and deterministic matrix sketching." Proceedings of the 19th ACM SIGKDD international conference on Knowledge discovery and data mining. 2013.

- Not clear if the Range-Net being "fully interpretable" is well-justified; I am not sure of the practical significance of this statement.

Additional Comments:
- The improvement in terms of memory requirements should be highlighted more in the description of the results (e.g., Section 4.3, where SketchySVD requires 24.91GB when only 4.19MB are required by RangeNet).

- Description of the exact methodology proposed (i.e., what are precisely the steps / optimization problems) should be highlighted more clearly in the main paper. Currently, a lot of the details are compressed as part of the Implementation Details subsection, and pushed into the Appendix which is not ideal.

- "It is well known that natural data matrices have a decaying spectrum wherein saving the data in memory in its original form is either redundant or not required from an application point of view." It'd be useful if the authors could provide with some concrete example use cases here.

- The "no loss in accuracy" statement in the last sentence of the first paragraph of the Introduction is quite unclear to me; any useful approximation through SVD for data matrices is by definition lossy.



**Summary Of The Paper:**

The paper presents a streaming method to compute an approximate Singular Value Decomposition, without requiring to load the entire data matrix into the main memory. The first stage of the method identifies a basis optimizing for the constraint of orthonormality, as well as minimizing the error between the input and the reconstruction of the input based on the learnt basis. The second stage corresponds to a rotation step, which aligns the low-rank approximation extracted with the SVD factors.

**Summary Of The Review:**

Strong results in terms of memory efficiency, which do justify consideration for acceptance in my view; however, the paper could improve in terms of clarity of exposing experiment setup details and choices of hyper-parameters, description of the main methodology, and having stronger application use cases.

---

> ### Author Response · Authors · 2021-11-17
> **Comments for Reviewer vVFi**
>
> 1. The extent to which the choice of the hyper-parameters...
>
> We have only shown results using Adamax as the gradient descent solver since our primary concern was to achieve lower errors with an error tolerance that the user can prescribe. The user can employ any of the gradient descent solvers (Adamax, Adagrad, etc.) rest assured that with epoch the error will reduce to the desired tolerance as per our theoretical results. Range-Net is guaranteed to reduce errors as the number of iterations increasing (shows convergence) since the loss non-convex loss surface has $2^r$ minima that are all equal energy (tail energy upon convergence). Please see Appendix C for details.
>
> We found Adamax to be the fastest in reaching machine precision errors in our numerical experiments. Note that none of the randomized-SVD solvers so far are able to achieve this (streaming or otherwise). Please see the supplementary code provided that shows the errors for a number of these latest randomized-solvers.
>
> 2. There is very limited discussion ... (e.g., Sandy Big Data ).
>
> We apologize for presenting some of the numerical experiment details in Appendices due to page limitation. The primary motivation of our work is that for a large dataset (like Sandy) one cannot perform full SVD on the entire dataset since the main memory requirement of a conventional SVD solver is prohibitive. SVD is used for exploratory analysis to identify at what spatial scale and time instance do vortices (turbulent features) occur and whether the vortices are attached to structure (locality in space) or detached (locality in time).
>
> 3. The authors could add more detail on why only SketchySVD was used as a baseline..
>
> We would like to clarify that the baseline solver is a conventional-SVD solver that requires the entire data matrix to be present in the main memory (RAM). Randomized-SVD algorithms have the same motivation: the data matrix is too large to be loaded into the main memory and as a pre-processing step a sketch (smaller matrix) of the original dataset is constructed on which conventional-SVD can be performed. The reviewer is referred to the supplementary code where a number of random-sketching methods are used to show that the errors in the computed SVD factors are large when compared to the baseline conventional SVD solver. Range-Net on the other hand is able to achieve machine precision errors against the same baseline conventional SVD solver.
>
> Thank you for suggesting this work and we have added the citation.
>
> 4. Not clear if the Range-Net being "fully interpretable" is well-justified
>
> The stage-1 and 2 learned weights are directly interpretable by the user as the right orthogonal matrix, and a unitary rotation matrix, respectively. This is to increase the user control over Range-Net computations. The user can easily verify Range-Net by substituting the right singular, rank-$r$ vector matrix as weights of stage-1 to find that the network loss is exactly the tail-energy as proposed by EYM.
>
> 5. The improvement in terms of memory requirements ...
>
> We appreciate this comment since going through the paper again we realized this was not highlighted sufficiently and that we focused too much on the errors.
>
> 6. Description of the exact methodology proposed
>
> We agree with the reviewer and can rearrange as suggested. However, we would really appreciate any inputs (page limitation) if some of the current sections in the main text can be moved to the appendix in exchange for the suggested details.
>
> 7. It is well known that natural data matrices have a decaying spectrum
>
> Please see the reference to Fig. 3 in the line following this sentence that shows compression as an application where the objective is to compress a given dataset while preserving a user-specified information content (as a percentage of variance in the original data). Also see the numerical experiment in Section 4.2 for MNIST feature dimension reduction, where the top $200$ eigenvectors $V$ are required for faithful inferences in any classification or clustering algorithm.
>
> 8. The "no loss in accuracy" statement ... SVD for data matrices is by definition lossy.
>
> We completely agree with the reviewer that any rank-$r$ approximation of a rank-$f$ matrix $X \in \mathbb{R}^{m \times n}$ where ($r<f$) is by definition lossy (approximation). More precisely, $X \neq X_{r}$ where $X_{r}$ is the rank-$r$ approximation of $X$. By no loss in accuracy we mean that the $X_{r,Range-Net}$ computed using Range-Net is close to $X_{r,Conventional-SVD}$ computed using a conventional full-SVD solver (baseline) with machine (GPU) precision absolute errors using Frobenius norm: $\|X_{r,Range-Net} - X_{r,Conventional-SVD}\|_{F}$. Range-Net not only has a lower main memory foot-print but also has substantially lower errors that can be controlled by a user specified tolerance if needed. Randomized-SVD errors on the other hand are irreducible (due to random sketching) in line with our theoretical result in Theorem 2.

---

> > ### Comment · Reviewer_vVFi · 2021-12-04
> > **Reviewer's response**
> >
> > Thanks to the authors for their response; I would like to acknowledge reading that. My score remains the same; I don't find that the authors' responses to concerns (e.g., related to baselines/experiments) would justify an increase in the score.

---

> > > ### Author Response · Authors · 2021-12-04
> > > **Thank you**
> > >
> > > We understand the score remaining unchanged. Irrespective of the score we hope the reviewer is convinced (given the code for a number of state of the art randomized methods) that irreducible errors are introduced due to sketching.

---

### Official Review · Reviewer_uKtm · 2021-11-08

**Correctness:** 3
**Technical Novelty And Significance:** 2
**Empirical Novelty And Significance:** 2
**Recommendation:** 5
**Confidence:** 2

**Main Review:**

The main selling point of the algorithm is that the memory needed is only O(rn), as opposed to O(r(m+n)) for competing algorithms and the result is exact as opposed to approximate.  The way this reduction is achieved is by allowing multiple streaming passes over the rows of X, whereas the competing algorithms only allow 1 pass.  This would be useful when the dataset is so large that storing a copy of the data is prohibitive and that is a realistic setting in the modern data-science workflow.

Indeed, for the purpose of dimensionality reduction it might even be sensible to only to complete stage 1.  That's because many algorithms, e.g. linear/logistic regression, kernel methods with the Euclidean kernel, and many neural network architectures, will be agnostic to the rotation H.

One small theoretical disadvantage is that the number of passes required is not known.  The authors state at most 5 passes are necessary, but actually that number was determined empirically and your number of passes depends on the number of epochs to convergence in stage 1.

Table 1 states that Range-Net space complexity is r*(n+r), however Appendix E.3 states that the implementation dumps XV*, an intermediate mxr matrix, to disk.  Specificly E.3 states "The output data [of stage 1] is dumped onto the secondary memory assuming that storing a low rank approximation is still main memory intensive. For Stage 2, this low rank approximation in the secondary memory is streamed as input, and the extracted singular values and vectors are saved in main memory." so the authors' implementation actually requires r*(m+n) storage."

I think it is misleading that the implementation does not obey the claimed memory bound.  It is true that the algorithm can be implemented without dumping XV*, but it requires more compute and passes to recalculate XV* during stage 2 and still more to compute the actual singular values.  How this extra compute affect the time required for the approximation?

In terms of writing, I think that the paper is poorly organized.  The main drawback is that there is no clear and concise description of the algorithm.  Parts of the algorithm are described in Section 3 and Appendix E, but Section 3 also contains the statements of correctness interspersed with the description.  There are terms left undefined and dimensions of matrices are only identified haphazardly.

Other notes:
. Please formally define tail energy.
. Section 1.2.  "In the absence of ... different from SVD factors".  I didn't understand this sentence at all and Equation (1) does not include any decomposition.
. I found the description of the algorithm as a neural network confusing. The network is two layers that each consist of a single matrix multiplication,  it's two single matrix multiplications which are solved for independently.
. Figure 4 caption does not match the subfigure captions



**Summary Of The Paper:**

The paper presents a multipass streaming algorithm for rank-r SVD.  Given an input matrix X in R^{mxn} the algorithm identifies two matrices V* in R^{nxr} and H in R^{rxr}. V* has orthonormal columns that span the top r right singular vectors.  H rotates V* so that V*H = Vr, where Vr in R^{nxr} is the matrix of the top-r right singular vectors.  V* and H are computed by minibatch gradient descent, trained until convergence with custom loss functions.  V* is computed in the first stage and then H in the second stage.

The algorithm is accurate and uses very little storage.

**Summary Of The Review:**

The algorithm is interesting and practical for is extremely small memory footprint, but organizational problems prevent a higher score.

---

> ### Author Response · Authors · 2021-11-17
> **Comments for Reviewer uKtm**
>
> 1. The main selling point of the algorithm is .... will be agnostic to the rotation H.
>
> We appreciate the reviewer comments and have addressed some of the concerns below.
>
> 2. One small theoretical disadvantage ... convergence in stage 1.
>
> We agree with the reviewer completely. Currently 5-passes (or epochs) is an empirical observation and not a theoretical result. However, note that even when using conventional SVD, the number of iterations cannot be guaranteed a-priori and depends on the spectral gap/ difference in the singular values inherent to the data. If the gap becomes smaller, the number of Range-Net passes must increase and is congruent with conventional SVD solvers. We would like to point out that with conventional SVD solvers, each iterations makes a complete pass over the entire data matrix and therefore requires that the data matrix be persistently be present in the main memory (RAM). Our approach does not require the data matrix to be persistently present in the main memory. Range-Net iterates over a subset of the data (as with any neural network) thereby reducing the main memory (RAM) load.
>
> 3. Table 1 states that Range-Net space complexity  ...  requires r*(m+n) storage."}
>
> We apologize for the confusion, the space complexity shown in the work considers peak main memory load (RAM). The choice of dumping $XV^*$ on to the secondary memory (HDD/SSD) lies with the user. The user can also choose not to perform the dumping step at all and stream the input data matrix completely from stage-1 to stage-2 of Range-Net. Therefore, the main-memory (RAM) load remains $r(n+r)$, independent of the number of samples $m$. Only the peak RAM load is shown in the numerical results. Note that for randomized-SVD approaches also, the memory load is with respect to the main memory (RAM) consumption and therefore the space complexity is also described in such approach with respect to it. We follow the same approach for a fair comparison of peak memory load against these methods.
>
> 4. I think it is misleading ... time required for the approximation?
>
> We have added the compute times in Table 2, 3 and 4 of main text. Note that a time-comparison between two algorithms solving the same problem is only fair if the same error tolerance can be prescribed for both. In other words, the time-comparison makes sense only when the two compared approaches generate equally accurate results. As shown in our work, for randomized-SVD methods the error is (1) dependent on the data matrix (2) is irreducible due to randomized sketching. Although, we have added our compute times for the numerical experiments please do note that our errors are at machine precision while randomized-SVD errors remain are 5-6 orders of magnitude larger. Please refer to the attached notebook in Supplemental material (code supplied) to confirm the errors (scale) in the singular vectors and values extracted for a few popular and recent Randomzied SVD methods.
>
> 5. In terms of writing .... identified haphazardly.}
>
> The proofs and theorems were added to the Appendix since the page limit did not allow us to completely present everything together. The two-stage network comprises of two non-convex optimization problems solved using gradient descent and is presented in Fig. 2. We have gone through the text to make sure that the dimensions are specified completely. However, we would really appreciate if the reviewer can point to any specific instance of such issues.
>
> 6. Other notes
>
> All the preliminaries and definitions are described in the appendix since the definition of tail-energy and associated terms were originally prescribed in the seminal work by Eckart-Young-Mirsky (EYM). We hope the Appendix was accessible in the supplementary material. We did not add the appendix in the main file due to ICLR's page limit restriction and desk reject policy.
>
> The method was described as a neural network since we intend to use it as an adaptive-SVD that can run in parallel with a larger neural-network (with a prescribed user-intent) to perform a run-time reduction of an arbitrary width and depth network. In other words, a low-memory SVD that can be performed simultaneously on a carrier network without interrupting or introducing a serial step it's training process.
>
> The reviewer is absolutely correct, that the stage-1 and stage-2 are two independent non-convex optimization problems that can either be solver separately or in a sequence simultaneously. We have left this choice to the user based upon their end objective.
>
> We thank the reviewer for pointing out the mismatch of captions in Figure 4 and have rectified them.

---

### Author Response · Authors · 2021-11-17
**Overall Comments**

Irrespective of the final decision, we would like to thank the reviewers for suggesting changes and indicating discussion areas.
1. We have added a easy-to use one-click-run Jupyter-notebook that will allow the reviewer to easily see the errors when using a number of different randomized SVD methods.
2. As shown in Theorem 2 (in main text and accompanied proof in Appendix), any projection matrix (random or otherwise) that is different from $V_{r}V_{r}^{T}$ ($V_{r}$ is the top rank-$r$ right singular vector matrix) will introduce irreducible errors in both singular values and vectors. This can be numerically verified using the well-known conventional SVD to baseline all other randomized SVD algorithms.
3. Our main intent in this work is to show that in addition to theoretical justification, numerical results directly supporting the theoretical arguments can be shown through benchmark problems. We therefore advocate that a common consensus should be reached as to how a proposed method can be shown to be better or worse compared to algorithms available in the literature.
4. Time-complexity and run-time comparisons between two algorithms can only be performed when the errors can be controlled and specified by the user (or tester) by prescribing a tolerance. Comparing run-times of two methods where the errors are scales apart is not a fair benchmark to declare a winner.
5. In our numerical experiments, we found that the relative tail energy as an error metric is misleading and smaller numbers here can still result in large deviation in the singular vectors and values or a low rank approximation. The user should be given control over these errors through a desired tolerance that satisfy their respective application area demands.

---

### Author Response · Authors · 2021-12-01
**Discussion period**

We did not receive any further comments after the rebuttal period. Is this normal for the review process?

---

> ### Comment · Area_Chair_DjDU · 2021-12-03
> **discussion**
>
> That is abnormal, and I'm working on getting some responses from the reviewers.  Sorry about that.
> -- AC

---

> > ### Author Response · Authors · 2021-12-03
> > **Thank you**
> >
> > As reviewers we completely understand that the number of submissions puts considerable load on the reviewers. We are thankful to you and the reviewers for looking into this.

---

### Decision · Program_Chairs · 2022-01-20

**Decision:**

Reject

**Comment:**

The reviewers were fairly consistent in agreeing that this is a reasonable paper with an interesting idea.  However, the use-case is fairly narrow, as the main benefit is less intermediate storage (and only significant for very rectangular matrices) but compared to alternatives it require many passes over the data (usually 5 or so). So it's a narrow use-case and many of the comparisons are not apples-to-apples since the accuracy, time, space-complexity and number of passes differ from algorithm to algorithm.

So while acknowledging the potential benefits of the method, there are downsides too, and thus a clear presentation is very essential. The reviewers mention that presentation (listing the algorithm, clear experiments) could be improved.

On my own reading, I noted that the choice of SketchySVD as the dominant baseline is misleading. SketchySVD is for streaming data (more restrictive than single-pass) so this is an unfair comparison. The appendix does a better job of including other baselines (block Lanczos), though it mischaracterizes them (it says "BlockLanczos requires persistence presence of the data matrix X in memory", but this is not true, the method could easily be implemented in a matrix-free fashion). Another method to compare with is the single-pass algorithm randSVD in Yu et al., who show how to implement one of the Halko et al. 2011 2-pass methods in just one-pass.  Other reviewers mention baseline algorithm issues too.  I do acknowledge the improved accuracy of your method over all these baselines for some matrices, in terms of the Frobenius norm (or tail error); however, I'm not sure the differences in spectral norm are are great, and see Remark 2.1 in Martinsson and Tropp '20 for arguments about why Frobenius norm guarantees are often not as desirable as spectral norm guarantees.

Another issue is related to the left vs right singular vectors. A reviewer noted: "It is not fair to compare RangeNet with SketchSVD, RangeNet just produces the right singular vectors while SketchSVD produces both left and right singular vectors." The authors respond "Range-Net computes both left and right singular vectors but does not consume main memory to store left singular vectors at run-time". However, if we allow another pass over the matrix to find the left singular vectors, this post-processing can be applied to *any* technique that approximates the singular values and right singular vectors, hence PCA methods are applicable, including deterministic methods like the "Frequent Directions" method (Ghashami et al. '16).

In summary, this method is high-accuracy and low-memory, yet it also has downsides compared to other methods, and the paper could use some improvement.  I don't think the paper is ready at this time for acceptance, but given the advantages of the method, I encourage the authors to make changes and resubmit an improved version to ICLR next year or other similar venue.


References:

Yu, Gu, Li, Liu, Li, "Single-Pass PCA of Large High-Dimensional Data". IJCAI '17, https://doi.org/10.24963/ijcai.2017/468

Ghashami, Liberty, Phillips, Woodruff, "Frequent directions: Simple and deterministic matrix sketching". SIAM Journal on Computing. 2016;45(5):1762-92.

Martinsson, Tropp. "Randomized numerical linear algebra: Foundations and algorithms". Acta Numerica. 2020 May;29:403-572.